# AED: Adaptable Error Detection
# for Few-shot Imitation Policy

**Jia-Fong Yeh**[1]     **Kuo-Han Hung**[1,*]     **Pang-Chi Lo**[1,*]     **Chi-Ming Chung**[1]
**Tsung-Han Wu**[2]     **Hung-Ting Su**[1]     **Yi-Ting Chen**[3]     **Winston H. Hsu**[1,4]
[1]National Taiwan University     [2]University of California, Berkeley
[3]National Yang Ming Chiao Tung University     [4]MobileDrive

## Abstract

We introduce a new task called Adaptable Error Detection (AED), which aims to identify behavior errors in few-shot imitation (FSI) policies based on visual observations in novel environments. The potential to cause serious damage to surrounding areas limits the application of FSI policies in real-world scenarios. Thus, a robust system is necessary to notify operators when FSI policies are inconsistent with the intent of demonstrations. This task introduces three challenges: (1) detecting behavior errors in novel environments, (2) identifying behavior errors that occur without revealing notable changes, and (3) lacking complete temporal information of the rollout due to the necessity of online detection. However, the existing benchmarks cannot support the development of AED because their tasks do not present all these challenges. To this end, we develop a cross-domain AED benchmark, consisting of 322 base and 153 novel environments. Additionally, we propose Pattern Observer (PrObe) to address these challenges. PrObe is equipped with a powerful pattern extractor and guided by novel learning objectives to parse discernible patterns in the policy feature representations of normal or error states. Through our comprehensive evaluation, PrObe demonstrates superior capability to detect errors arising from a wide range of FSI policies, consistently surpassing strong baselines. Moreover, we conduct detailed ablations and a pilot study on error correction to validate the effectiveness of the proposed architecture design and the practicality of the AED task, respectively. The AED project page can be found at `https://aed-neurips.github.io/`.

## 1   Introduction

Few-shot imitation (FSI), a framework that learns a policy in novel (unseen) environments from a few demonstrations, has recently drawn significant attention in the community [1, 2, 3, 4, 5, 6]. Notably, the framework, as exemplified by [7, 8, 9, 10, 1, 11, 2], has demonstrated its efficacy across a range of robotic manipulation tasks. This framework shows significant potential to adapt to a new task based on just a few demonstrations from their owners [1, 2]. However, a major barrier that still limits their ability to infiltrate our everyday lives is the ability to detect behavior errors in novel environments.

We propose a challenging and crucial task called adaptable error detection (AED), aiming to monitor FSI policies from visual observations and report their behavior errors, along with the corresponding benchmark. In this work, behavior errors refer to states that deviate from the demonstrated behavior, necessitating the timely termination of the policy upon their occurrence. Unlike existing few-shot visual perception tasks [12], failures can result in significant disruptions to surrounding objects and humans in the real world. This nature often restricts real-world experiments to simple tasks. Our AED benchmark is built within Coppeliasim [13] and Pyrep [14], encompassing six indoor tasks and one factory task. This comprehensive benchmark comprises 322 base and 153 new environments,

38th Conference on Neural Information Processing Systems (NeurIPS 2024).

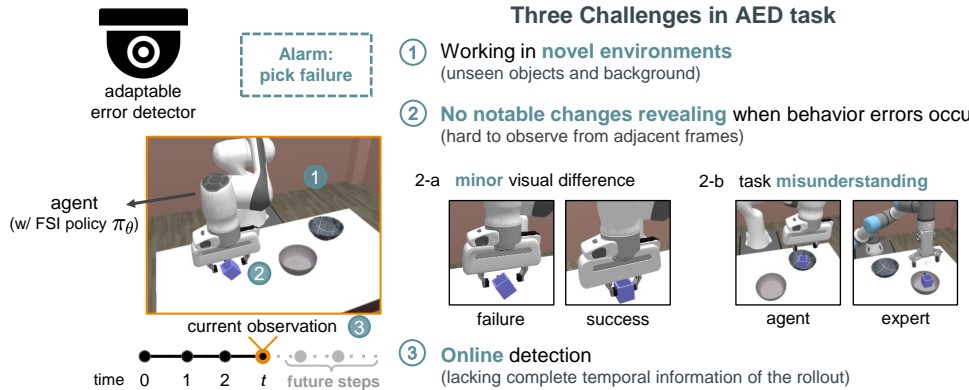

Figure 1: Our novel adaptable error detection (AED) task. To monitor the behavior of the few-shot imitation (FSI) policy $\pi_\theta$, the adaptable error detector needs to address three challenges: (1) it works in novel environments, (2) no notable changes reveal when behavior errors occur, and (3) it requires online detection. These challenges make existing error detection methods infeasible.

spanning diverse domains and incorporating multiple stages. We aim to create a thorough evaluation platform for AED methodologies, ensuring their effectiveness before real-world deployment.

The AED task present three novel challenges, illustrated in Figure 1. First, AED entails monitoring a policy's behavior in novel environments, the normal states of which are not observed during training. Second, detecting behavior errors becomes challenging, as there are no noticeable changes to indicate when such errors occur. Specifically, this makes it difficult to discern either minor visual differences or misunderstandings of the task through adjacent frames. Third, AED requires online detection to terminate the policy timely, lacking complete temporal information of the rollout.

Current approaches struggle to address the unique challenges posed by AED. One-class classification (OCC) methods, including one-class SVMs [15] or autoencoders [16, 17, 18, 19], face difficulties in handling unseen environments. These methods are trained solely with normal samples and identify anomalies by evaluating their significant deviation from the learned distribution. However, normal states in novel environments, which include unseen backgrounds and objects, are already considered out-of-distribution for these techniques, resulting in subpar performance. Multiple instance learning [20] or patch-based methods [21, 22] may alleviate the second challenge, particularly minor visual differences, in seen environments. However, the feasibility of applying these methods to AED remains underexplored. Video few-shot anomaly detection (vFSAD) methods [21, 23, 24] are unsuitable for AED due to their dependency on full temporal observation from videos.

To this end, we introduce Pattern Observer (PrObe), a novel algorithm that extracts discriminative features from the monitored policy to identify instances of behavior errors. Specifically, PrObe designs a gating mechanism to extract task-relevant features from the monitored FSI policy to mitigate the impact of a novel environment **(first challenge)**. Then, we design an effective loss function to distill sparse pattern features, making it easier to observe changes in observation **(second challenge)**. Additionally, we design a recurrent generator to generate a pattern flow of current policy. Finally, we determine if there is a behavior error by proposing a novel temporal-aware contrastive objective to compare the pattern flow and demonstrations **(third challenge)**.

We conduct thorough experiments on the proposed benchmark. Even when faced with various policy behaviors and different characteristics of strong baselines, our PrObe still achieved the highest Top 1 counts, average ranking, and average performance difference (with a maximum difference of up to 40%), demonstrating its superiority. Additionally, we conducted an extensive ablative study to justify the effectiveness of our design choices. Furthermore, we reported additional experimental results covering timing accuracy, embedding visualization, demonstration quality, viewpoint changes, and **error correction** to validate our claims and the practicality of our task and method.

**Contributions**   Our work makes three significant contributions: (1) We define a vital yet under-explored task called Adaptable Error Detection (AED) and develop its associated benchmark to facilitate collective exploration by the research community. (2) We introduce PrObe, a novel method

that monitors the policy's behavior by retrieving patterns from its feature embeddings. (3) We conduct thorough evaluations on the proposed benchmark, demonstrating the effectiveness of PrObe. It surpasses several baselines and shows robustness across different policies. We anticipate that our research will serve as a key foundation for future real-world experiments in the field of FSI research.

## 2 Related Work

### 2.1 Few-shot Imitation (FSI)

**Policy**   With the recent progress in meta-learning [25], the community explores the paradigm for learning policies from limited demonstrations during inference [26, 27, 28]. Notably, these works either assume that the agent and expert have the same configuration or explicitly learn a motion mapping between them [27]. Conversely, DC methods [1, 11, 29] develop a policy that behaves conditioned both on the current state and demonstrations. Furthermore, they implicitly learn the mapping between agents and experts, making fine-tuning optional. Thereby, effectively extracting knowledge from demonstrations becomes the most critical matter. In most FSI works, no environment interactions are allowed before inference. Hence, policies are usually trained by behavior cloning (BC) objectives, i.e., learning the likelihood of expert actions by giving expert observations. Recently, DCRL [1] trains a model using reinforcement learning (RL) objects and performs FSI tasks without interacting with novel environments.

**Evaluation tasks**   Since humans can perform complex long-horizon tasks after watching a few demonstrations, FSI studies continuously pursue solving long-horizon tasks to verify if machines can achieve the same level. A set of research applies policy on a robot arm to perform daily life tasks in the real world [27] or simulation [7]. The task is usually multi-stage and composed of primitives/skills/stages [11, 2], such as a typical pick-and-place task [28] or multiple boxes stacking [7]. Besides, MoMaRT [19] tackles a challenging mobile robot task in a kitchen scene.

Our work formulates the AED task for the safety concern of FSI policies and proposes PrObe to address it, which is valuable for extensive FSI research. Besides, we have also built challenging FSI tasks containing attributes such as scenes from different fields, realistic simulation, task distractors, and various robot behaviors

### 2.2 Few-shot Anomaly Detection (FSAD)

**Problem setting**   Most existing FSAD studies deal with anomalies in images [22, 30, 31, 32, 33] and a few tackle anomalies in videos [34, 35, 36, 24]. Moreover, problem settings are diverse. Some works presume only normal data are given during training [30, 33, 34, 35], while others train models with normal and a few anomaly data and include unknown anomaly classes during inference [31, 36].

**Method summary**   Studies that only use normal training samples usually develop a reconstruction-based model with auxiliary objectives, such as meta-learning [34], optical flow [35], and adversarial training [30, 35]. Besides, patch-based methods [31, 22] reveal the performance superiority on main image FSAD benchmark [37] since the anomaly are tiny defeats. Regarding video few-shot anomaly detection (vFSAD), existing works access a complete video to compute the temporal information for determining if it contains anomalies. In addition, vFSAD benchmarks [38, 39] provide frame-level labels to evaluate the accuracy of locating anomalies in videos.

**Comparison between vFSAD and AED task**   Although both the vFSAD and our AED task require methods to perform in unseen environments, there are differences: (1) The anomalies in vFSAD involve common objects, while AED methods monitor the policy's behavior errors. (2) An anomaly in vFSAD results in a notable change in the perception field, such as a cyclist suddenly appearing on the sidewalk [38]. However, no notable change is evident when a behavior error occurs in AED. (3) The whole video is accessible in vFSAD [21, 23, 24], allowing for the leverage of its statistical information. In contrast, AED requires online detection to terminate the policy timely, lacking the complete temporal information of the rollout. These differences make our AED task more challenging and also render vFSAD methods infeasible.

# 3 Preliminaries

**Few-shot imitation (FSI)** FSI is a framework worthy of attention, accelerating the development of various robot applications. Following [11], a FSI task is associated with a set of base environments $E^b$ and novel environments $E^n$. In each novel environment $e^n \in E^n$, a few demonstrations $\mathcal{D}^n$ are given. The objective is to seek a policy $\pi$ that achieves the best performance (e.g., success rate) in novel environments leveraging a few demonstrations. Note that, a task in base and novel environments are semantically similar, but their backgrounds and interactive objects are disjoint. The framework takes as input $N$ demonstrations (collected by a RGB-D camera) and an RGB-D image of the current observation, following the setting of [10, 11]. Addressing FSI tasks typically involves three challenges in practice [11]: (1) The task is long-horizon and multi-stage. (2) The demonstrations are length-variant, making each step misaligned, and (3) The expert and agent have a distinct appearance or configuration. Developing a policy to solve the FSI task and simultaneously tackle these challenges is crucial.

**Demonstration-conditioned (DC) policy** As stated above, the expert and agent usually have different appearances or configurations. The DC policy $\pi(a \mid s, \mathcal{D})$ learns an implicit mapping using current states $s$ and demonstrations $\mathcal{D}$ to compute agent actions $a$. Next, we present the unified architecture of DC policies and how they produce the action. When the observations $o$ and demonstrations are RGB-D images that only provide partial information, we assume that the current history $h_t := (o_1, o_2, ..., o_t)$ and demonstrations $\mathcal{D}$ are adopted as inputs.

A DC policy comprises a feature encoder, a task-embedding network, and an actor. After receiving the rollout history $h$, the feature encoder extracts the history features $f_h$. Meanwhile, the feature encoder also extracts the demonstration features. Then, the task-embedding network computes the task-embedding $f_\zeta$ to retrieve task guidance. Notably, the lengths of agent rollout and demonstrations can vary. The task-embedding network is expected to handle length-variant sequences by padding frames to a prefixed length or applying attention mechanisms. Afterward, the actor predicts the action for the latest observation, conditioned on the history features $f_h$ and task-embedding $f_\zeta$. Additionally, an optional inverse dynamics module predicts the action between consecutive observations to improve the policy's understanding of how actions affect environmental changes. At last, the predicted actions are optimized by the negative log-likelihood or regression objectives (MSE).

# 4 Adaptable Error Detection (AED)

Our AED task is formulated to monitor and report FSI policies' behavior errors, i.e., states of policy rollouts that are inconsistent with demonstrations. The challenges posed by the AED task have been presented in Figure 1. We formally state the task and describe the protocol we utilized below.

**Task statement** Let $c$ denote the category of agent's behavior status, where $c \in \mathbf{C}, \mathbf{C} = \{\text{normal}, \text{error}\}$. When the agent with the trained FSI policy $\pi_\theta$ performs in a novel environment $e^n$, an adaptable error detector $\phi$ can access the agent's rollout history $h$ and a few expert demonstrations $\mathcal{D}^n$. It then predicts the categorical probability $\hat{y}$ of the behavior status for the latest state in the history $h$ by $\hat{y} = \phi(h, \mathcal{D}^n) = P(c \mid enc(h, \mathcal{D}^n))$, where $enc$ denotes the feature encoder, and it may be $enc_\phi$ ($\phi$ contains its own encoder) or $enc_{\pi_\theta}$ (based on policy's encoder). Next, let $y$ represent the ground truth probability, we evaluate $\phi$ via the expectation of detection accuracy over agent rollouts $X^n$ in all novel environments $E^n$:

$$\mathbb{E}_{e^n \sim E^n} \mathbb{E}_{e^n, \pi_\theta(\cdot \mid \cdot, \mathcal{D}^n)} \mathbb{E}_{h \sim X^n} A(\hat{y}, y), \tag{1}$$

where $A(\cdot, \cdot)$ is the accuracy function that returns 1 if $\hat{y}$ is consistent with $y$ and 0 otherwise. However, frame-level labels are often lacking in novel environments due to the established assumption [8, 11] that we have less control over them. Therefore, we employ a sequence-level $A(\cdot, \cdot)$ in our experiments.

**Protocol** We explain the utilized protocol (shown in Figure 2) with a practical scenario: A company develops a home robot assistant. This robot can perform a set of everyday missions in customers' houses (novel environments $E^n$), given a few demonstrations $\mathcal{D}^n$ from the customer. Before selling, the robot is trained in the base environments $E^b$ built by the company. In this scenario, both the

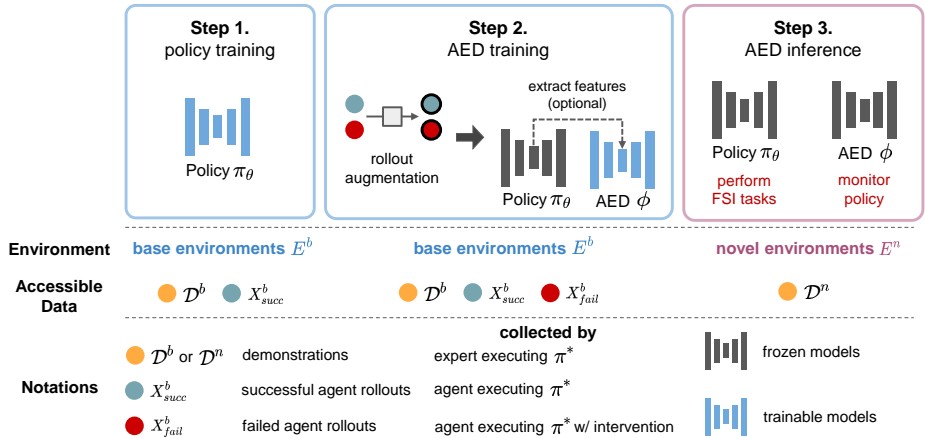

Figure 2: Our AED protocol. the successful agent rollouts $X_{succ}^b$, failed agent rollouts $X_{fail}^b$, and a few expert demonstrations $\mathcal{D}^b$ are available for all base environments $E^b$. Then, the task contains three phases: policy training, AED training, and AED inference. We aim to train an adaptable error detector $\phi$ to report policy $\pi_\theta$'s behavior errors when performing in novel environments $E^n$.

agent's and expert's optimal policies $\pi^*$ are available during training, with an established assumption from the FSI society [8, 10, 11] that base environments are highly controllable. Then, we can collect successful agent rollouts $X_{succ}^b$ and a few expert demonstrations $\mathcal{D}^b$ for all base environments [1]. Next, we also collect failed agent rollouts $X_{fail}^b$ by intervening in the agent's $\pi^*$ at a critical timing (e.g., the moment to grasp objects) so that $X_{fail}^b$ can possess precise frame-level error labels.

With these resources, our utilized AED protocol consists of three phases: (1) The policy $\pi_\theta$ is optimized using successful agent rollouts $X_{succ}^b$ and a few demonstrations $\mathcal{D}^b$. (2) The adaptable error detector $\phi$ is trained on agent rollouts $X_{succ}^b$, $X_{fail}^b$ and a few demonstrations $\mathcal{D}^b$. Besides, the detector $\phi$ may use features extracted from policy $\pi_\theta$'s encoder, whose parameters are not updated in this phase. (3) The policy $\pi_\theta$ solves the task leveraging few demonstrations $\mathcal{D}^n$, and the detector $\phi$ monitors the policy's behavior simultaneously. Notably, no agent rollouts are collected in this phase. Only a few demonstrations $\mathcal{D}^n$ are available since the agent is now in novel environments $E^n$.

## 5 Pattern Observer (PrObe)

**Overview** To address the AED task, we develop a rollout augmentation approach and a tailored AED method. The rollout augmentation aims to increase the diversity of collected rollouts and prevent AED methods from being overly sensitive to subtle differences in rollouts. Regarding the AED method, our insight is to detect behavior errors from policy features that contain task-related knowledge, rather than independently training an encoder to judge from visual observations alone. Thus, we propose Pattern Observer (PrObe), which discovers the unexpressed patterns in the policy features extracted from frames of successful or failed states. Even if the visual inputs vary during inference, PrObe leverages the pattern flow and a consistency comparison to effectively alleviate the challenges posed by the AED task.

### 5.1 Rollout Augmentation

To ensure a balanced number of successful and failed rollouts (i.e., $X_{succ}^b$ and $X_{fail}^b$), along with precise frame-level labels, we gather them using the agent's optimal policy $\pi^*$ (with intervention). However, even if the policy $\pi_\theta$ is trained on $X_{succ}^b$, it inevitably diverges from the agent's optimal policy $\pi^*$ due to limited rollout diversity. Therefore, AED methods trained solely on the raw collected agent rollouts will be too sensitive to any subtle differences in the trained policy's rollouts, leading to high false positive rates.

---

[1]The successful agent rollouts are not expert demonstrations since they might have different configurations.

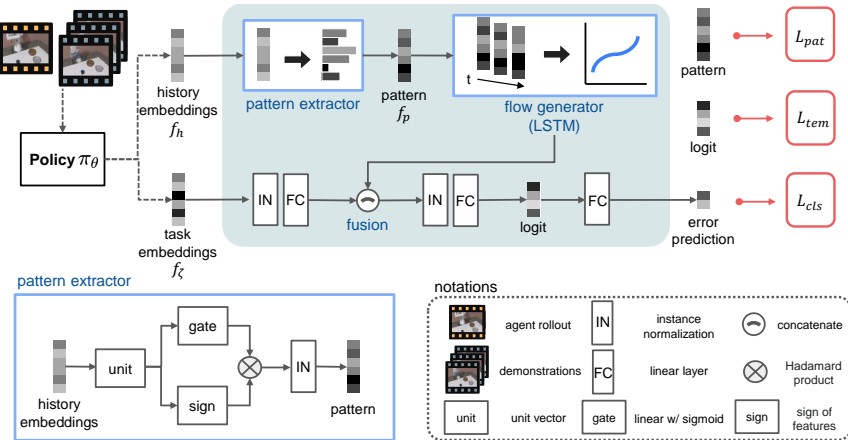

Figure 3: Architecture of PrObe. PrObe detects behavior errors through the pattern extracted from policy features. The learnable gated pattern extractor and flow generator (LSTM) compute the pattern flow of history features $f_h$. Then, the fusion with transformed task-embeddings $f_\varsigma$ aims to compare the task consistency. PrObe predicts the behavior error based on the fused embeddings. Objectives, $L_{pat}$, $L_{tem}$, and $L_{cls}$, optimize the corresponding outputs.

Accordingly, we augment agent rollouts so as to increase their diversity and dilute the behavior discrepancy between trained $\pi_\theta$ and agent's $\pi^*$. Specifically, we iterate through each frame and its label from sampled agent rollouts and randomly apply the following operations: *keep*, *drop*, *swap*, and *copy*, with probabilities of 0.3, 0.3, 0.2, and 0.2, respectively. This process injects distinct behaviors into the collected rollouts, such as speeding up (interval frames dropped), slowing down (repetitive frames), and non-smooth movements (swapped frames). We demonstrate that this approach can contribute to AED methods, as shown in Figure 11 in the Appendix.

## 5.2 PrObe Architecture

As depicted in Figure 3, PrObe comprises three major components: a pattern extractor, a pattern flow generator, and an embedding fusion. First, the pattern extractor take as input the history features $f_h$ from the trained policy $\pi_\theta$, aiming to retrieve discriminative information from each embedding of $f_h$. Precisely, the pattern extractor transforms $f_h$ into unit embeddings through division by its L2 norm, thus mitigating biases caused by changes in visual inputs (**first challenge**). Then, a learnable gate composed of a linear layer with a sigmoid function determines the importance of each embedding cell. A Hadamard product between the sign of unit embeddings and the importance scores, followed by instance normalization (IN), is applied to obtain the pattern features $f_p$ for each timestep.

Second, PrObe feeds $f_p$ into an LSTM (flow generator, **third challenge**) to generate the pattern flow. Intuitively, the changes in the pattern flow of successful and failed rollouts will differ. On the other hand, the task-embeddings $f_\varsigma$ extracted from $\pi_\theta$ are transformed by an IN layer, a linear layer (FC), and a tanh function, mapping the task-embeddings into a space similar to the pattern flow.

Third, a fusion process concatenates the pattern flow and transformed task-embeddings to compute the error predictions $\hat{y}$. This process is expected to compare the consistency between the agent rollout and demonstrations and uses this as a basis for determining whether behavior errors occur.

**Objective Functions**  PrObe is optimized by one supervised and two unsupervised objectives. Firstly, a binary cross-entropy objective $L_{cls}$ is employed to optimize the error prediction $\hat{y}$, as frame-level labels $y$ are available during training. Secondly, the L1 loss $L_{pat}$ is applied to the pattern features $f_p$, encouraging the pattern extractor to generate sparse pattern features, facilitating the observation of pattern changes (**second challenge**). Thirdly, a contrastive objective $L_{tem}$, a temporal-aware variant of triplet loss [40], is developed and applied to the logit embeddings to emphasize the difference between the normal and error states in failed rollouts $X_{fail}^b$ (**third challenge**). The idea behind $L_{tem}$ is that adjacent frames' logits contain similar signals due to the temporal information from the pattern flow, even in the presence of errors (cf. Figure 7 in the Appendix). Blindly pushing

the logits of normal and error states far apart could mislead the model and have adverse effects. Therefore, $L_{tem}$ considers the temporal relationship between samples and is calculated as follows:

$$L_{tem} = \frac{1}{N} \sum_{i=0}^{N} \max(\|\text{logit}_i^a - \text{logit}_i^p\|_2 - \|\text{logit}_i^a - \text{logit}_i^n\|_2 + m_t, 0). \tag{2}$$

Here, $N$ is the number of sampled pairs, and the temporal-aware margin $m_t = m * (1.0 - \alpha * (d_{ap} - d_{an}))$ adjusts based on the temporal distance of anchor, positive and negative samples. $m$ represents the margin in the original triplet loss, while $d_{ap}$, $d_{an}$ are the clipped temporal distances between the anchor and positive sample, and the anchor and negative sample, respectively. Ultimately, PrObe is optimized through a weighted combination of these three objectives.

## 6 Experiments

Our experiments seek to address the following questions: (1) Is it better to solve the AED task by analyzing discernible patterns in policy features rather than using independent encoders to extract features from visual observations? (2) How do our architecture designs contribute, and how do they perform in various practical situations? (3) Can our AED task be effectively integrated with error correction tasks to provide greater overall contribution?

### 6.1 Experimental Settings

**Evaluation tasks**  Existing manipulation benchmarks [41, 42, 43, 44] cannot be used to evaluate the AED task because they do not effectively represent all challenges posed by the AED task, such as cross-environment training and deployment. Therefore, we have designed seven cross-domain and multi-stage robot manipulation tasks that encompass both the challenges encountered in FSI [11] and our AED task. Detailed task information is provided in **Section D of the Appendix**, including mission descriptions, schematics, configurations, and possible behavioral errors. Our tasks are developed using Coppeliasim [13], with Pyrep [14] serving as the coding interface.

**FSI policies**  To assess the ability of AED methods to handle various policy behaviors, we implement three demonstration-conditioned (DC) policies to perform FSI tasks, following the descriptions in Section 3 and utilizing the same feature extractor and actor architecture. The primary difference among these policies lies in how they extract task embeddings from expert demonstrations. NaiveDC [8][2] concatenates the first and last frames of demonstrations and averages their embeddings to obtain task-embeddings; DCT [1][3] employs cross-demonstration attention and fuses them at the time dimension; SCAN [11] computes tasks-embeddings using stage-conscious attention to identify critical frames in demonstrations. More details can be found in **Section A of the Appendix**.

**Baselines**  In our newly proposed AED task, we compare PrObe with four strong baselines, each possessing different characteristics, as detailed in **Section C of the Appendix**. Unlike PrObe, all baselines incorporate their own encoder to distinguish errors. We describe their strategies for detecting errors here: (1) **SVDDED**: A deep one-class SVM [16] determines whether the current observation deviates from the centroid of demonstration frames (*single frame, OOD*). (2) **TaskEmbED**: A model [8] distinguishes the consistency between the current observation and demonstrations (*single frame, metric-learning*). (3) **LSTMED**: A deep LSTM [45] predicts errors solely based on current rollout history (*temporal*). (4) **DCTED**: A deep transformer [1] distinguishes the consistency between the current rollout history and demonstrations (*temporal, metric-learning*).

**Metrics**  We report *the area under the receiver operating characteristic* (AUROC) and *the area under the precision-recall curve* (AUPRC), two conventional threshold-independent metrics in error/anomaly detection literature [46, 47]. To compute these scores, we linearly select 5000 thresholds spaced from 0 to 1 (or SVDDED's outputs) and apply a sequence-level accuracy function (cf. **Section E.1 in the Appendix**). Furthermore, we conduct an evaluation to verify if AED methods can identify behavior errors timely, as depicted in Figure 5.

---

[2]TaskEmb → NaiveDC, avoiding confusion w/ AED baselines.

[3]DCRL → DCT (transformer), since we don't use RL training.

| Close Drawer | SVDDED | TaskEmbED | LSTMED | DCTED | PrObe |
|---|---|---|---|---|---|
| NaiveDC (91.11%) | 0.7404 | 0.8395 | 0.8186 | 0.8250 | **0.9133** |
| | 0.2626 | 0.6840 | 0.7700 | 0.7813 | **0.8350** |
| DCT (80.56%) | **0.8378** | 0.7081 | 0.6590 | 0.6413 | 0.7680 |
| | 0.7039 | 0.5995 | 0.6493 | 0.6739 | **0.7438** |
| SCAN (88.33%) | 0.4079 | 0.6498 | **0.7867** | 0.7218 | 0.6978 |
| | 0.1411 | 0.2220 | **0.6677** | 0.6390 | 0.5829 |

| Press Button | SVDDED | TaskEmbED | LSTMED | DCTED | PrObe |
|---|---|---|---|---|---|
| NaiveDC (51.94%) | 0.4113 | 0.4872 | 0.3769 | 0.7194 | **0.7957** |
| | 0.6956 | 0.7280 | 0.7051 | 0.8405 | **0.8851** |
| DCT (80.56%) | 0.5710 | 0.5429 | 0.4886 | 0.7306 | **0.7506** |
| | 0.3789 | 0.3597 | 0.4468 | 0.5734 | **0.7474** |
| SCAN (75.56%) | 0.4280 | 0.4754 | 0.4066 | 0.7491 | **0.7782** |
| | 0.4653 | 0.4657 | 0.4487 | 0.6757 | **0.7505** |

| Pick & Place | SVDDED | TaskEmbED | LSTMED | DCTED | PrObe |
|---|---|---|---|---|---|
| NaiveDC (55.00%) | 0.7074 | 0.6810 | 0.7116 | 0.7493 | **0.7635** |
| | 0.7971 | 0.8028 | 0.8148 | 0.8041 | **0.8665** |
| DCT (64.05%) | 0.6125 | 0.7523 | 0.6555 | 0.7743 | **0.8173** |
| | 0.7001 | 0.8173 | 0.7304 | 0.8294 | **0.8780** |
| SCAN (71.19%) | 0.5646 | 0.6381 | 0.7050 | 0.7482 | **0.8012** |
| | 0.5078 | 0.6065 | 0.7103 | 0.7063 | **0.8072** |

| Move Glass Cup | SVDDED | TaskEmbED | LSTMED | DCTED | PrObe |
|---|---|---|---|---|---|
| NaiveDC (42.25%) | 0.4772 | 0.7907 | 0.7152 | 0.6884 | **0.8684** |
| | 0.7837 | 0.9368 | 0.8961 | 0.9028 | **0.9605** |
| DCT (88.00%) | 0.5515 | 0.7975 | 0.7354 | 0.6635 | **0.8342** |
| | 0.3181 | 0.6968 | 0.5314 | 0.3608 | **0.7961** |
| SCAN (58.25%) | 0.4749 | 0.7879 | 0.7660 | 0.6548 | **0.8046** |
| | 0.6292 | 0.8820 | 0.8704 | 0.7354 | **0.8980** |

| Organize Table | SVDDED | TaskEmbED | LSTMED | DCTED | PrObe |
|---|---|---|---|---|---|
| NaiveDC (12.20%) | 0.2581 | **0.8078** | 0.5946 | 0.5844 | 0.6808 |
| | 0.8911 | **0.9786** | 0.9611 | 0.9452 | 0.9652 |
| DCT (79.00%) | 0.5622 | 0.6382 | **0.7001** | 0.6702 | 0.6550 |
| | 0.4800 | 0.3922 | **0.6352** | 0.5515 | 0.5905 |
| SCAN (66.60%) | 0.5000 | 0.6193 | 0.6734 | 0.5962 | **0.6759** |
| | 0.5086 | 0.4865 | 0.6356 | 0.5255 | **0.6983** |

| Back to Box | SVDDED | TaskEmbED | LSTMED | DCTED | PrObe |
|---|---|---|---|---|---|
| NaiveDC (08.89%) | 0.4621 | 0.3221 | 0.4148 | 0.6022 | **0.8446** |
| | 0.9367 | 0.9375 | 0.9555 | 0.9679 | **0.9903** |
| DCT (29.17%) | 0.5537 | 0.7220 | **0.7707** | 0.4772 | 0.7411 |
| | 0.8652 | 0.9314 | **0.9546** | 0.8534 | 0.9438 |
| SCAN (58.89%) | 0.5498 | 0.7041 | 0.8237 | 0.7489 | **0.8614** |
| | 0.6433 | 0.7401 | 0.8913 | 0.7880 | **0.9097** |

| Factory Packing | SVDDED | TaskEmbED | LSTMED | DCTED | PrObe |
|---|---|---|---|---|---|
| NaiveDC (45.42%) | 0.3338 | 0.5564 | 0.6471 | 0.5361 | **0.6635** |
| | 0.5583 | 0.7057 | 0.7667 | 0.6772 | **0.7676** |
| DCT (88.75%) | 0.3849 | 0.7201 | **0.7934** | 0.7509 | 0.7670 |
| | 0.1600 | 0.5002 | 0.6335 | 0.6224 | **0.6759** |
| SCAN (63.75%) | 0.6151 | 0.6916 | 0.7836 | 0.5006 | **0.8256** |
| | 0.5657 | 0.7287 | 0.8063 | 0.6306 | **0.8622** |

| Statistics | SVDDED | TaskEmbED | LSTMED | DCTED | PrObe |
|---|---|---|---|---|---|
| Top 1 counts | 1 | 1 | 4 | 0 | **15** |
| | 0 | 1 | 3 | 0 | **17** |
| Avg. ranking | 4.4 | 3.2 | 2.9 | 3.1 | **1.4** |
| | 4.6 | 3.5 | 2.7 | 3.0 | **1.2** |
| Avg. difference | 7.05% | 41.63% | 41.59% | 44.20% | **67.29%** |
| | 2.61% | 35.42% | 61.67% | 57.13% | **78.03%** |

Figure 4: Performance comparison of AED methods on seven challenging FSI tasks. The values under each policy indicate its success rate for each task. AUROC[↑] and AUPRC[↑] scores are listed in the upper and lower rows for each policy, respectively, ranging from 0 to 1. According to the statistics table, PrObe achieves the highest Top 1 counts (15 and 17 out of 21 cases), average ranking, and average performance difference in both metrics, demonstrating its superiority and robustness.

## 6.2 Analysis of Experimental Results

**Main experiment - detecting policies' behavior errors**   Our main experiment aims to verify whether discerning policies' behavior errors from their features is an effective strategy. We follow the protocol in Figure 2 to conduct the experiment. Specifically, the FSI policies perform all seven tasks, and their rollouts are collected. Then, the AED methods infer the error predictions for these rollouts. Lastly, we report two metrics based on their inference results. Notably, the results of policy rollout are often biased (e.g., many successes or failures). Thus, AUPRC provides more meaningful insights in this experiment because it is less affected by the class imbalance of outcomes.

Due to the diverse nature of policy behaviors and the varying characteristics of baselines, achieving consistently superior results across all tasks is highly challenging. Nonetheless, according to Figure 4, our PrObe achieved the highest Top 1 counts, average ranking, and average performance difference in both metrics, indicating PrObe's superiority among AED methods and its robustness across different policies. We attribute this to PrObe's design, which effectively addresses the challenges in the AED task. The average performance difference is the average performance gain compared to the worst method across cases, calculated as $(\text{score} - \text{worst\_score})/\text{worst\_score}$.

Additionally, we investigated cases where PrObe exhibited suboptimal performance, which can be grouped into two types: (1) Errors occurring at specific timings (e.g., at the end of the rollout) and identifiable without reference to the demonstrations (e.g., DCT policy in *Organize Table*). (2) Errors that are less obvious compared to the demonstrations, such as when the drawer is closed with a small gap (SCAN and DCT policies in *Close Drawer*). In these cases, the pattern flow does not change enough to be recognized, resulting in suboptimal results for PrObe.

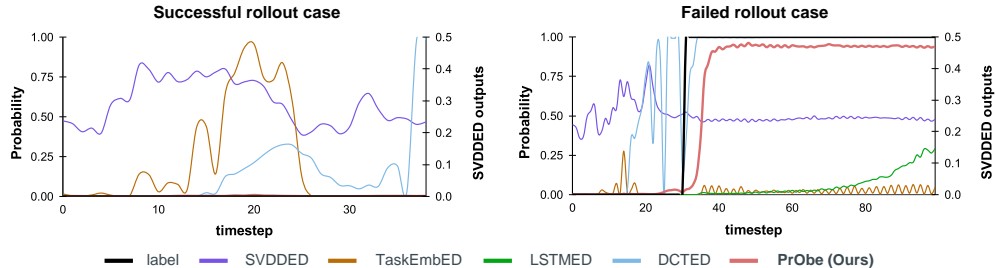

Figure 5: Visualization of timing accuracy. Raw probabilities and SVDDED outputs of selected successful (left) and failed (right) rollouts are drawn. PrObe raises the error at the accurate timing in the failed rollout and stably recognizes normal states in the successful case.

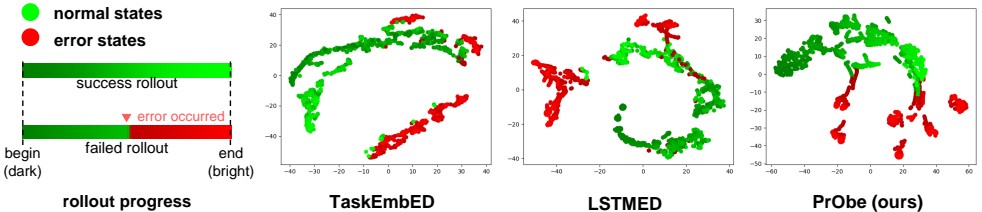

Figure 6: t-SNE visualization of learned embeddings (representations). The green and red circles represent normal and error states, respectively. The brightness of the circle indicates the rollout progress (from dark to bright). The embeddings learned by our PrObe have better interpretability because they exhibit task progress and homogeneous state clustering characteristics.

**Timing accuracy** Our main experiment examines whether AED methods can accurately identify behavioral errors when provided with complete policy rollouts. To further validate the timing of their predictions, we annotated a subset of rollouts (specifically the SCAN policy in *Pick & Place*) and visualized the raw outputs of all AED methods. SVDDED outputs the embedding distance between observation and task-embedding, while others compute probabilities, as depicted in Figure 5.

In the successful rollout case, both our PrObe and LSTMED consistently identify states as normal, while other methods misidentify them, increasing the probability that inputs represent error states. Conversely, in the failed rollout case, PrObe detects errors after a brief delay ($< 5$ frames), significantly elevating the probability. We attribute this short delay to the time it takes for pattern flow to induce sufficient change. Nonetheless, PrObe detects errors with the closest timing; other methods either raise probabilities too early or too late. We emphasize that this phenomenon is common and not a deliberate choice.

**Embedding visualization** To analyze whether the learned embeddings possess discernible patterns, we initially extract the same 20 rollouts from the annotated dataset above using three AED methods to obtain the embeddings. Subsequently, we present the t-SNE transform [48] on these embeddings in Figure 6. Obviously, the embeddings learned by TaskEmbED and LSTMED are scattered and lack an explainable structure. In contrast, our PrObe's embeddings exhibit characteristics of task progress and homogeneous state clustering, i.e., the states with similar properties (e.g., the beginnings of rollouts, the same type of failures) are closer. This experiment supports the hypothesis that PrObe can learn implicit patterns from the policy's feature embeddings.

**Ablations and supportive experiments** In response to the second question, we summarize comprehensive ablations in the Appendix. First, Figure 11 indicates that the proposed **rollout augmentation (RA)** strategy increases the rollout diversity and benefits AED methods with temporal information. Second, Figure 12 demonstrates that **PrObe's design** effectively improves performance. Third, Figure 13 illustrates PrObe's **performance stability** by executing multiple experiment runs on a subset of tasks and computing the performance variance. Fourth, we examine how AED methods' performance is influenced when receiving **demonstrations with distinct qualities**, as presented in Table 7. Lastly, we study the influence of **viewpoint changes** in Table 8. Please refer to the corresponding paragraphs in the Appendix for exhaustive versions.

Table 1: Error correction results. **Success Rate (SR)** [↑] is reported. The detection threshold is set as 0.9 for all methods. The values indicate the performance of the DCT policy without correction (first column) and its collaboration with the correction policy and four AED methods (remaining columns). PrObe is the only method that improves the performance, as it detects the error most accurately.

| DCT policy | w/ TaskEmbED | w/ LSTMED | w/ DCTED | w/ PrObe |
|---|---|---|---|---|
| $80.56 \pm 11.65\%$ | $80.56 \pm 11.65\%$ | $80.28 \pm 11.60\%$ | $71.67 \pm 20.75\%$ | $\mathbf{82.22} \pm \mathbf{10.17}\%$ |

### 6.3 Pilot Study on Error Correction

To further examine the practicality of our AED task (the third question), we conducted a pilot study on error correction. In this experiment, the FSI policy is paused after the AED methods detect an error. Then, a correction policy from [19], which resets the agent to a predefined safe pose, is applied. Finally, the FSI policy continues to complete the task. We conducted the study on the *Press Button* task, where errors are most likely to occur and be corrected. Besides, the correction policy is defined as moving to the top center of the table. We allowed the DCT policy to cooperate with the correction policy and four AED methods (SVDDED excluded), as summarized in Table 1.

We have two following findings: (1) Our PrObe is verified to be the most accurate method once again. In contrast, other AED methods may cause errors at the wrong timing, making it challenging for the correction policy to improve performance (it may even have a negative impact on original successful trials). (2) The performance gain from the correction policy is minor, as it lacks precise guidance in unseen environments.

We believe that error correction in novel environments warrants a separate study due to its challenging nature, as observed in the pilot study. One potential solution is the human-in-the-loop correction, which operates through instructions [49, 50] or physical guidance [51]. However, their generalization ability and cost when applying to our AED task need further discussion and verification. We will leave this as a topic for our future work.

## 7 Conclusion

We point out the importance of monitoring policy behavior errors to accelerate the development of FSI research. To this end, we formulate the novel adaptable error detection (AED) task, whose three challenges make previous error detection methods infeasible. To address AED, we propose the novel Pattern Observer (PrObe) by detecting errors in the space of policy features. With the extracted discernible patterns and additional task-related knowledge, PrObe effectively alleviates AED's challenges. It achieves the best performance, as confirmed by both our primary experiment and thorough ablation analyses. We also demonstrate PrObe's robustness in the timing accuracy experiment and the learned embedding visualization. Additionally, we provide a pilot study on error correction, revealing the practicality of the AED task. Our work is an essential cornerstone in developing future FSI research to conduct complex real-world experiments.

**Limitations and future work**   We carefully discuss the limitations of our work in Section F of the Appendix, including online AED inference and real-world experiments. For future work, developing a unified AED method applicable to various tasks and policies is worth exploring. Additionally, discovering previously unseen erroneous behaviors remains an interesting and challenging avenue.

## Acknowledgements

This work was supported in part by the National Science and Technology Council, Taiwan, under Grant NSTC 112-2634-F-002-006. We are grateful to MobileDrive and the National Center for High-Performance Computing. We also thank all reviewers and area chairs for their valuable comments and positive recognition of our work during the review process.

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

# Appendix

This appendix presents more details of our work, following a similar flow to the main paper. First, Section A introduces the few-shot imitation (FSI) policies. Then, our PrObe's details are provided in Section B. Besides, the strong AED baselines are depicted in Section C. Next, the designed multi-stage FSI tasks are exhaustively described in Section D, including task configurations, design motivations, and possible behavior errors. In addition, more experimental results are shown in Section E to support our claims. At last, we responsibly present our work's limitations in Section F.

## A  FSI Policy Details

We follow the main paper's Section 3 to implement three state-of-the-art demonstration-conditioned (DC) policies, including NaiveDC[8], DCT[1], and SCAN[11], and monitor their behaviors.

Table 2: General components' settings of FSI policies

| Visual head | |
|---|---|
| input shape | $T \times 4 \times 120 \times 160$ |
| architecture | a resnet18 with a FC layer and a dropout layer |
| out dim. of resnet18 | 512 |
| out dim. of visual head | 256 |
| dropout rate | 0.5 |
| Actor | |
| input dim. | 512 or 768 |
| architecture | a MLP with two parallel FC layers (pos & control) |
| hidden dims. of MLP | [512 or 768, 256, 64] |
| out dim. of pos layer | 3 |
| out dim. of control layer | 1 |

**Architecture settings**  The same visual head and actor architecture are employed for all policies to conduct a fair comparison. Table 2 and Table 3 present the settings of general components (visual head and actor) and respective task-embedding network settings for better reproducibility.

In the visual head, we added a fully-connected and a dropout layer after a single ResNet18 to mitigate overfitting of the data from base environments. Moreover, the input dimension of the actor module is determined by adding the dimensions of the outputs from the visual head and the task-embedding network. For the DCT and SCAN policies, the task-embedding's dimension matches the visual head's output. However, the task-embedding's dimension in NaiveDC policy is twice the size.

Regarding the task-embedding network, NaiveDC concatenates the first and last frames of each demonstration Subsequently, it computes the average as the task-embedding, without involving any attention mechanism; DCT incorporates cross-demonstration attention to fuse demonstration features, followed by applying rollout-demonstration attention to compute the task-embedding; SCAN utilizes an attention mechanism to attend to each demonstration from the rollout, aiming to filter out uninformative frames. It then fuses attended frames to obtain the final task-embedding. Furthermore, we employ a standard LSTM in our implementation, differing from the setting described in [11].

**Training details**  To optimize the policies, we utilize a RMSProp optimizer [52] with a learning rate of 1e-4 and a L2 regularizer with a weight of 1e-2. Each training epoch involves iterating through all base environments. Within each iteration, we sample five demonstrations and ten rollouts from the sampled base environment. The total number of epochs varies depending on the specific tasks and is specified in Table 5. All policy experiments are conducted on a Ubuntu 20.04 machine equipped with an Intel i9-9900K CPU, 64GB RAM, and a NVIDIA RTX 3090 24GB GPU.

## B  PrObe Details

We propose the Pattern Observer (PrObe) to address the novel AED task by detecting behavior errors from the policy's feature representations, which offers two advantages: (1) PrObe has a better

Table 3: Task-embedding network settings of FSI policies

| NaiveDC | |
|---|---|
| input data | demonstrations |
| process | concat the first and last demonstration frames and average |
| out dim. | 512 |

| DCT (Transformer-based) | |
|---|---|
| input data | current rollout and demonstrations |
| process | a cross-demo attention followed by a rollout-demo attention |
| number of encoder layers | 2 |
| number of decoder layers | 2 |
| number of heads | 8 |
| normalization eps | 1e-5 |
| dropout rate | 0.1 |
| out dim. | 256 |

| SCAN | |
|---|---|
| input data | current rollout and demonstrations |
| process | rollout-demo attentions for each demonstration and then average |
| number of LSTM layer | 1 |
| bi-directional LSTM | False |
| out dim. | 256 |

understanding of the policy because the trained policy remains the same during both AED training and testing. This consistency enhances its effectiveness in identifying errors. (2) The additional task knowledge from the policy encoder aids in enhancing the error detection performance. We have described PrObe's architecture and their usages in Section 5 of the main paper. Here, we introduce more PrObe's design justification and training objectives.

**PrObe's design justification** The pattern extractor aims to extract observable patterns from embeddings in policy features. One potential approach is to leverage discrete encoding, as used in [53]. However, the required dimension of discrete embedding may vary and be sensitive to the evaluation tasks. To address variability, we leverage an alternative approach that operates in a continuous but sparse space. Additionally, the goal of the pattern flow generator is to capture the temporal information of current patterns. Due to the characteristics of the AED task, a standard LSTM model is better suited than a modern transformer. This is because policy rollouts are collected sequentially, and adjacent frames contain crucial information, especially when behavior errors occur. Finally, PrObe detects inconsistencies by comparing the pattern flow with the transformed task embedding. The contributions of PrObe's components are verified and shown in Figure 12. The results demonstrate that our designed components significantly improve performance.

**Objectives** We leverage three objectives to guide our PrObe: a BCE loss $L_{cls}$ for error classification, a L1 loss $L_{pat}$ for pattern enhancement, and a novel temporal-aware triplet loss $L_{tem}$ for logit discernibility. First, the PrObe's error prediction $\hat{y}$ can be optimized by $L_{cls}$ since the ground-truth frame-level labels $y$ are accessible during training, which is calculated by,

$$L_{cls} = -\frac{1}{N_r}\frac{1}{T_h}\sum_{i=0}^{N_r}\sum_{j=0}^{T_h}(y_{i,j} \cdot \ln \hat{y}_{i,j} + (1 - y_{i,j}) \cdot \ln(1 - \hat{y}_{i,j})), \qquad (3)$$

where $N_r$ and $T_h$ are the number and length of rollouts, respectively. Then, we leverage the L1 objective $L_{pat}$ to encourage the pattern extractor to learn a more sparse pattern embedding $f_p$, which can be formulated by

$$L_{pat} = \frac{1}{N_r}\frac{1}{T_h}\sum_{i=0}^{N_r}\sum_{j=0}^{T_h}|f_{p,i,j}|. \qquad (4)$$

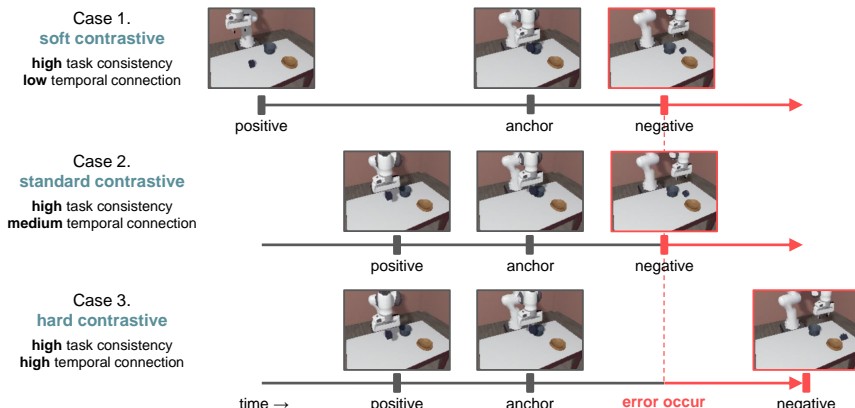

Figure 7: Schematic of our novel temporal-aware triplet loss $L_{tem}$. When the anchor and positive sample are more related in both the time and task aspects, the temporal-aware margin $m_t$ will be larger (the closer the two embeddings are encouraged). Besides, $m_t$ provides a slight encouragement in case the positive sample and anchor are far away on temporal distance.

With the objective $L_{pat}$, the pattern embeddings are expected to be sparse and easily observable of changes, benefiting the model in distinguishing behavior errors.

Next, we highlight that the time relationship should also be considered when applying contrastive learning to temporal-sequence data, in addition to determining whether the frames are normal or errors for the task. Thus, $L_{tem}$ is a novel temporal-aware triplet loss [40], and the temporal-aware margin $m_t$ in $L_{tem}$ will have different effects depending on the relationship between the anchor and the positive/negative sample, as depicted in Figure 7. The $L_{tem}$ can be calculated by:

$$L_{tem} = \frac{1}{N} \sum_{i=0}^{N} \max(\|\text{logit}_i^a - \text{logit}_i^p\|_2 - \|\text{logit}_i^a - \text{logit}_i^n\|_2 + m_t, 0), \quad (5)$$

where $N$ is the number of sampled pairs, and the temporal-aware margin $m_t = m * (1.0 - \alpha * (d_{ap} - d_{an}))$ will be enlarged or reduced considering the temporal distance of anchor, positive and negative samples. Among them, $m$ is the margin in the original triplet loss, and $d_{ap}$, $d_{an}$ are the clipped temporal distances between the anchor and positive sample and the anchor and negative sample, respectively. With $L_{tem}$, PrObe can efficiently perform contrastive learning without getting confused by blindly pulling temporally distant anchors and positive samples closer.

Lastly, the total loss $L_{total}$ is the combination of three objectives:

$$L_{total} = \lambda_{pat} L_{pat} + \lambda_{tem} L_{tem} + \lambda_{cls} L_{cls}, \quad (6)$$

where $\lambda_{pat}$, $\lambda_{tem}$, and $\lambda_{pat}$ are weights to balance different objectives.

Table 4: Attributes of error detection methods

| method name | policy dependent | training rollouts | temporal information | DC-based | output type | # of parameters |
|---|---|---|---|---|---|---|
| SVDDED | | only successful | | ✓ | distance | 11.31M |
| TaskEmbED | | both | | ✓ | probability | 11.52M |
| LSTMED | | both | ✓ | | probability | 11.85M |
| DCTED | | both | ✓ | ✓ | probability | 14.49M |
| PrObe (ours) | ✓ | both | ✓ | ✓ | probability | 0.65M |

## C Error Detector Details

We compare our PrObe with several strong baselines that possesses different attributes, aiming to verify their effectiveness on addressing the adaptable error detection (AED) task. In this section, we comprehensively present the attributes and training details associated with these baselines.

**Baseline attributes** Table 4 presents the attributes of AED baselines and our PrObe. All baselines have their own encoder and are independent of the policies, which offers the advantage that they only need to be trained once and can be used for various policies. However, as a result, they recognize errors only based on visual information. Now we describe their details:

- **SVDDED**: A modified deep one-class SVM method [16] trained only with successful rollouts. It relies on measuring the distance between rollout embeddings to the center embedding to detect behavior errors, without considering the temporal information. We compute the center embedding by averaging demonstration features and minimize the embedding distance between rollout features and the center during training.

- **TaskEmbED**: A single-frame baseline trained with successful and failed rollouts, which is a modification from the NaiveDC policy [8]. It concatenates and averages the first and last demonstration frames as the task-embedding. Then, it predicts the behavior errors conditioned on the current frame and task-embedding.

- **LSTMED**: A deep LSTM model [45] predicts errors based solely on the current rollout. It is trained with successful and failed rollouts without access to demonstration data. It is expected to excel in identifying errors that occur at similar timings to those observed during training. However, it may struggle to recognize errors that deviate from the demonstrations.

- **DCTED**: A baseline with temporal information derived from the DCT policy [1]. It is trained with successful and failed rollouts and incorporates with demonstration information. One notable distinction between DCTED and our PrObe lies in their policy dependencies. DCTED detects errors using its own encoder, which solely leverages visual information obtained within the novel environment. Conversely, PrObe learns within the policy feature space, incorporating additional task-related knowledge.

**Training details** Similar to optimizing the policies, we utilize a RMSProp optimizer with a learning rate of 1e-4 and a weight regularizer with a weight of 1e-2 to train the error detection models. During each iteration within an epoch, we sample five demonstrations, ten successful agent rollouts, and ten failed rollouts from the sampled base environment for all error detection models except SVDDED. For SVDDED, we sample five demonstrations and twenty successful agent rollouts since it is trained solely on normal samples. Notably, all error detection experiments are conducted on the same machine as the policy experiments, ensuring consistency between the two sets of experiments.

## D Evaluation Tasks

To obtain accurate frame-level error labels in the base environments and to create a simulation environment that closely resembles real-world scenarios, we determined that existing robot manipulation benchmarks/tasks [54, 41, 42, 55, 11, 43, 44] did not meet our requirements. Consequently, seven challenging FSI tasks containing 322 base and 153 novel environments are developed. Their general attributes are introduced in Section D.1 and Table 5; the detailed task descriptions and visualizations are provided in Section D.2 and Figure 8-9, respectively.

### D.1 General Task Attributes

- **Generalization**: To present the characteristics of changeable visual signals in real scenes, we build dozens of novel environments for each task, comprising various targets and task-related or environmental distractors. Also, dozens of base environments are built as training resources, enhancing AED methods' generalization ability. Letting them train in base environments with different domains can also be regarded as a **domain randomization** technique [9], preprocessing for future sim2real experiments.

Table 5: FSI task configurations

| Task | # of stages | # of epochs | # of base env. | # of novel env. | # of task distractors | # of env. distractors |
|---|---|---|---|---|---|---|
| *Close Drawer* | 1 | 160 | 18 | 18 | 1 | 3 |
| *Press Button* | 1 | 40 | 27 | 18 | 0 | 3 |
| *Pick & Place* | 2 | 50 | 90 | 42 | 1 | 0 |
| *Move Glass Cup* | 2 | 50 | 42 | 20 | 1 | 0 |
| *Organize Table* | 3 | 50 | 49 | 25 | 1 | 0 |
| *Back to Box* | 3 | 50 | 60 | 18 | 0 | 3 |
| *Factory Packing* | 4 | 80 | 36 | 12 | 0 | 0 |

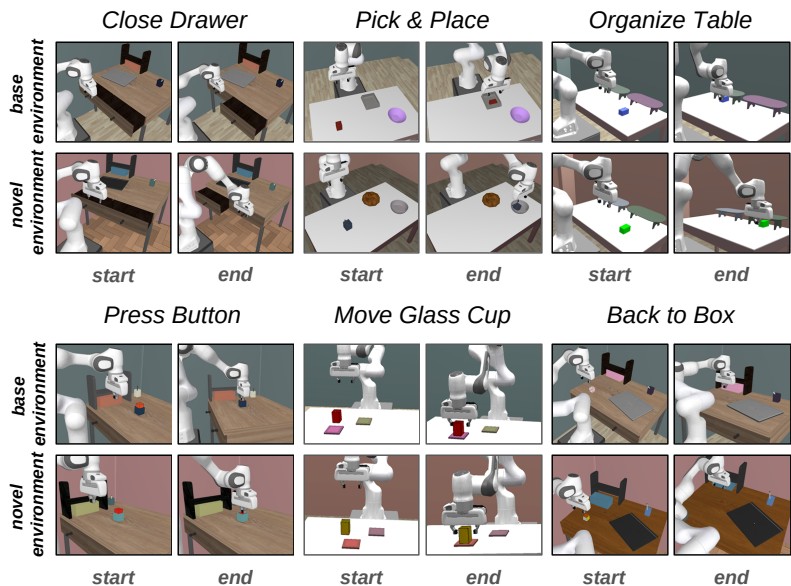

Figure 8: Visualizations of our designed indoor FSI tasks. The backgrounds (wall, floor) and interactive objects are disjoint between base and novel environments.

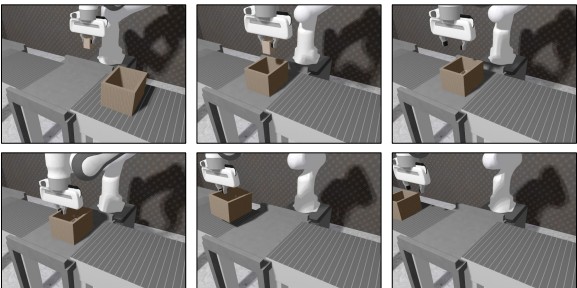

Figure 9: Progress visualization of our *Factory Packing* task. The agent needs to wait for the box's arrival. It then places the product into the box and lets the box move to the next conveyor belt.

- **Challenging**: Three challenges, including multi-stage tasks, misaligned demonstrations, and different appearances between expert and agent, are included in our FSI tasks. Moreover, unlike existing benchmarks (e.g., RLBench [41]) attaching the objects to the robot while grasping, we turn on the gravity and friction during the whole process of simulation, so grasped objects may drop due to unsmooth movement.

- **Realistic**: Our tasks support multi-source lighting, soft shadow, and complex object texture to make them closer to reality (cf. Figure 9). Existing benchmarks usually have no shadow (or only a hard shadow) and simple texture. Besides, all objects are given reasonable weights. The aforementioned gravity and friction also make our missions more realistic.

## D.2 Task Descriptions

| *Close Drawer (1-stage, indoor)* | |
| --- | --- |
| **Description** | Fully close the drawer that was closed by the expert in demonstrations. Besides, the objects on the table will be randomly set in the designated area. |
| **Success condition** | The target drawer is fully closed, and another drawer (distractor) must not be moved. |
| **Behavior errors** | (1) Not fully close the target drawer; (2) Close the distractor, not the target drawer; (3) Close two drawers at the same time. |

| *Press Button (1-stage, indoor)* | |
| --- | --- |
| **Description** | Fully press the button which is randomly placed in a designated area. |
| **Success condition** | The button is fully pressed. |
| **Behavior errors** | (1) Not touching the button at all; (2) The button is not fully pressed. |

| *Pick & Place (2-stages, indoor)* | |
| --- | --- |
| **Description** | Pick and place the mug/cup into the bowl the expert placed in demonstrations. |
| **Success condition** | The mug/cup is fully placed in the target bowl. |
| **Behavior errors** | (1) Failed to pick up the mug/cup; (2) Failed to place it into the target bowl; (3) Misplaced the mug/cup into another bowl (distractor). |

| *Move Glass Cup (2-stages, indoor)* | |
| --- | --- |
| **Description** | Pick up and place a glass of water on the coaster the expert placed in demonstrations. |
| **Success condition** | The glass of water is placed on the target coaster and no water is spilled. |
| **Behavior errors** | (1) Failed to pick up the glass of water; (2) drips spilling; (3) Not placed on the target coaster. |

| *Organize Table (3-stages, indoor)* | |
| --- | --- |
| **Description** | Pick up and place the object in front of the target bookshelf, and push it under the shelf. |
| **Success condition** | The object is fully under the target shelf. |
| **Behavior errors** | (1) Failed to pick up the object; (2) Losing object during moving; (3) Not fully placed under the target shelf. |

| *Back to Box (3-stages, indoor)* | |
| --- | --- |
| **Description** | Pick up the magic cube/dice and place it into the storage box. Then, push the box until it is fully under the bookshelf. |
| **Success condition** | The magic cube/dice is in the storage box, and the box is fully under the bookshelf. |
| **Behavior errors** | (1) Failed to pick up the magic cube/dice. (2) Failed to place the magic cube/dice into the storage box. (3) The box is not fully under the bookshelf. |

| *Factory Packing (4-stages, factory)* | |
| --- | --- |
| **Description** | Wait until the product box reaches the operating table, place the product in the box, and pick and place the box on the next conveyor belt. |
| **Success condition** | The product is inside the box, and the box reaches the destination table. |
| **Behavior errors** | (1) Failed to place the product into the box. (2) Failed to pick up the box. (3) Failed to place the box on the next conveyor belt. |

# E  Additional Experimental Results

This section reports more experimental results. We introduce the accuracy function used for determining an error prediction result in Section E.1. Furthermore, more details on the main experiment are provided in Section E.2, Next, all ablations are summarized in Section E.3. At last, the pilot study on error correction is shown in Section 6.3.

## E.1  Accuracy Function

| rollout successful? | any error raised? | marked as |
|:---:|:---:|:---:|
| ✓ | ✓ | false positive (FP) |
| ✓ | ✗ | true negative (TN) |
| ✗ | ✓ | true positive (TP) |
| ✗ | ✗ | false negative (FN) |

Figure 10: Rules in the accuracy function $A(\cdot, \cdot)$ to determine the error prediction results.

As stated in the main paper, our control and understanding are diminished in novel environments. Consequently, during inference, we lack frame-level labels, only discerning success or failure at the end of the rollout. We adopt similar rules from [19] to define the accuracy function $A(\cdot, \cdot)$, as depicted in Figure 10. This function determines the outcome of error prediction at the sequence-level, which may not adequately reflect the accuracy of timing for error detection. Hence, we proceed to conduct the timing accuracy experiment (Figure 5 in the main paper) to address this concern.

## E.2  Main Experiment's Details

Table 6: FSI policy performance. **Success rate (SR)** [↑] and its standard deviation (STD) are reported. STD is calculated across **novel environments**, rather than multiple experiment rounds. This accounts for the variability in STDs, as the difficulty of novel environments within the same task can vary.

| | Close Drawer | Press Button | Pick & Place | Move Glass Cup |
|---|---|---|---|---|
| NaiveDC | **91.11 ± 20.85**% | 51.94 ± 18.79% | 55.00 ± 24.98% | 42.25 ± 34.04% |
| DCT | 80.56 ± 26.71% | **80.56 ± 11.65**% | 64.05 ± 19.77% | **88.00 ± 14.87**% |
| SCAN | 88.33 ± 13.94% | 75.56 ± 12.68% | **71.19 ± 15.38**% | 58.25 ± 20.33% |

| | Organize Table | Back to Box | Factory Packing | |
|---|---|---|---|---|
| NaiveDC | 12.20 ± 13.93% | 08.89 ± 09.21% | 45.42 ± 30.92% | |
| DCT | **79.00 ± 09.27**% | 29.17 ± 09.46% | **88.75 ± 07.11**% | |
| SCAN | 66.60 ± 23.18% | **58.89 ± 10.87**% | 63.75 ± 33.11% | |

**FSI policy performance**  The policy performance is originally reported in Figure 4 of the main paper. Here, we provide a comprehensive overview in Table 6. Each policy conducts 20 executions (rollouts) for each novel environment to obtain the success rate (SR). Subsequently, the average success rate and its standard deviation (STD) are calculated across these SRs. Large STD values may arise due to the diversity in difficulty among novel environments. Factors such as variations in object size or challenging destination locations can lead to divergent performance outcomes for the policies. Additionally, these rollouts are collected for later AED inference.

In general, the NaiveDC policy achieves the best results in simple tasks because the task-embedding from the concatenation of the first and last two frames provides sufficient information. However, its performance rapidly declines as task difficulty increases. Besides, the DCT policy excels in tasks where the misalignment between demonstrations is less severe, as it fuses the task-embeddings in the temporal dimension. Lastly, the SCAN policy outperforms in tasks with high temporal variability because its attention mechanism filters out uninformative demonstration frames effectively.

**Main experiment's ROC curves**    We have reported detailed numbers from the main experiment in Figure 4 of the main paper. Here, Figure 15 (on page 25) presents the ROC curves of all tasks. The numbers reported in Figure 4 represent the area under the curves. To reiterate, our proposed PrObe achieves the best performance on both metrics and can handle the various behaviors exhibited by different policies.

### E.3   Ablation Study

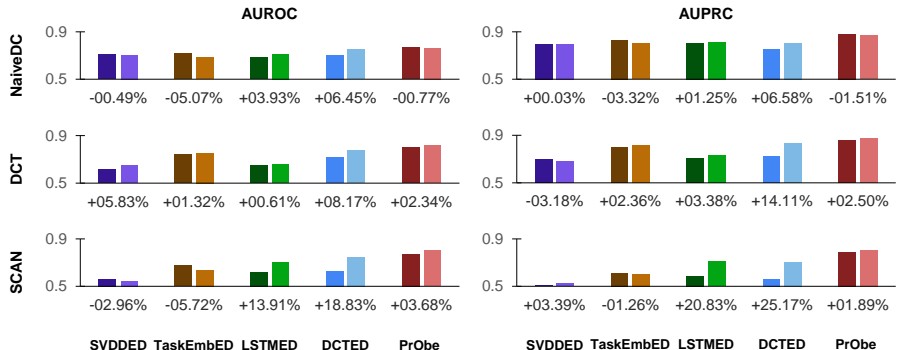

Figure 11: Effectiveness of rollout augmentation (RA). **AUROC**[↑] and **AUPRC**[↑] are reported. Methods trained without RA (dark) and with RA(bright) and their performance gap are listed. RA is more beneficial for methods with time information (rightmost three columns). Also, the improvement from RA is more evident when the performance of FSI policy is higher (SCAN case).

**Rollout augmentation**    This experiment aims to validate whether rollout augmentation (RA) enhances the collected agent rollout diversity and, consequently, improves the robustness of error detectors. Figure 11 presents the results of all error detection methods (trained with and without RA) monitoring policies solving the *Pick & Place* task, yielding the following findings: (1) RA has a minor or even negative impact on SVDDED, as expected. Since SVDDED is exclusively trained on single frames from successful rollouts, the removal of frames by RA decreases overall frame diversity. (2) RA proves beneficial for methods incorporating temporal information (LSTMED, DCTED, and PrObe), particularly when the FSI policy performs better (SCAN case). This is because error detectors cannot casually trigger an error, or they risk making a false-positive prediction. The results indicate that methods trained with RA generate fewer false-positive predictions, showcasing improved robustness to various policy behaviors.

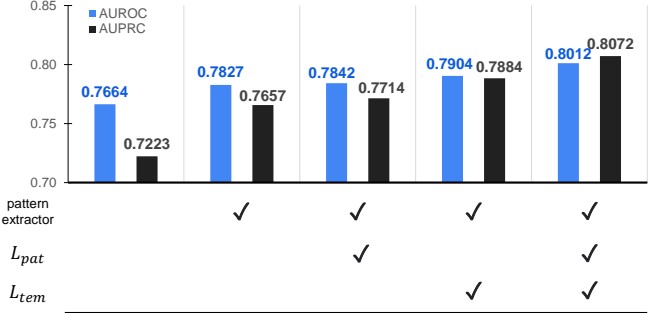

Figure 12: Component contributions. **AUROC**[↑] and **AUPRC**[↑] are reported. The pattern extractor and two auxiliary objectives significantly improve the performance.

**PrObe's design verification**    Figure 12 illustrates the performance contribution of each component of PrObe, observed while monitoring the SCAN policy solving the *Pick & Place* task. The naive model (first column) removes the pattern extractor and utilizes a FC layer followed by an IN layer to transform history embeddings. Clearly, the designed components and objectives enhance performance, particularly with the addition of the pattern extractor into the model. We attribute this improvement to the fact that the extracted embedding patterns by our PrObe are more informative and discernible.

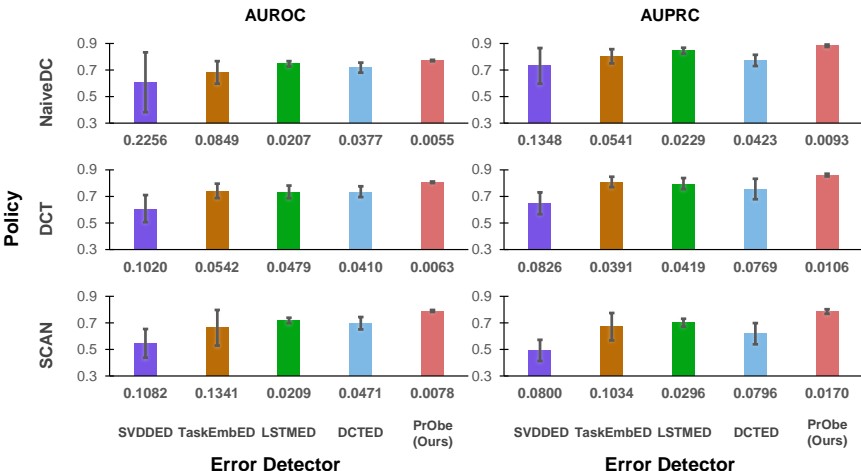

Figure 13: Performance results with error bars (standard deviation, STD). **AUROC**[↑], **AUPRC**[↑] are reported. Here, STDs are computed across **multiple experiment rounds** (random seeds). It is obvious that in the representative *Pick & Place* task, PrObe not only achieves the highest scores when detecting all policies' behavior errors but also has the best stability (the smallest STD).

**Performance stability**    Our work involves training and testing all policies and AED methods to produce results for an FSI task. Additionally, each task comprises numerous base and novel environments. These factors make it time-consuming and impractical to include error bars in the main experiment (Figure 4 of the main paper). Based on our experience, generating the results of the main experiment once would require over 150 hours.

Therefore, we selected the most representative *Pick & Place* task among our FSI tasks and conducted multiple training and inference iterations for all AED methods. We present the AUROC, AUPRC, and their standard deviations (STD) averaged over multiple experimental rounds (with five random seeds) in Figure 13. From the results, our proposed PrObe not only achieves the best performance in detecting behavior errors across all policies but also exhibits the most stable performance among all AED methods, with the smallest STD values.

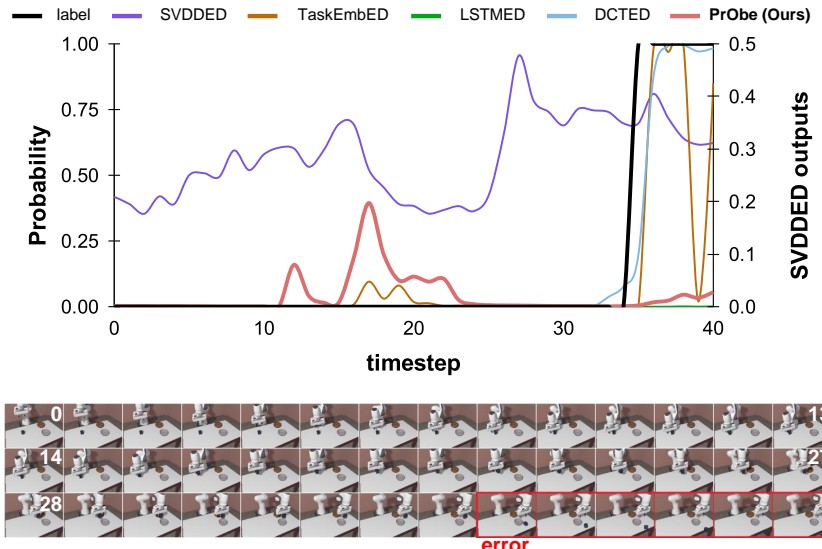

Figure 14: PrObe's failure prediction. In the *Pick & Place* task, the rollout is terminated immediately if the mug is no longer on the table, giving the error detectors only six frames to detect the error in this case. However, it took a while for our Probe's pattern flow to induce enough change. By the last frame, PrObe was just beginning to increase its predicted probability, but it was too late.

**PrObe's failure prediction**    We analyze cases in which PrObe is ineffective at detecting behavior errors and visualize the results in Figure 14. As stated in the main paper, our PrObe requires a short period after the error occurs to allow the pattern flow to induce enough changes. However, in the case depicted in Figure14, the mug had fallen off the table, resulting in the immediate termination of the rollout. Such a short duration is insufficient for PrObe to adequately reflect, thereby rendering it unable to substantially increase the prediction probability of behavioral errors at the last moment. Developing mechanisms for quick responsive pattern flow is one of our future research directions.

Table 7: Demonstration quality experiment. **AUPRC** [↑] is reported. This experiment verifies the influence of demonstration quality on both the FSI policy and AED methods. PrObe is the only method that can achieve higher performance when receiving sub-optimal demonstrations, as the FSI policies raise more failures.

| | NaiveDC policy | | | | | |
|---|---|---|---|---|---|---|
| Setting | Success Rate | SVDDED | TaskEmbED | LSTMED | DCTED | PrObe |
| all optimal | $51.94 \pm 18.79\%$ | 0.6956 | 0.7280 | 0.7051 | 0.8405 | 0.8851 |
| 3 optimal & 2 sub | $50.83 \pm 18.50\%$ | 0.6861 | 0.7407 | 0.7168 | 0.8832 | 0.9059 |
| all sub-optimal | $51.67 \pm 17.24\%$ | 0.6414 | 0.7276 | 0.7044 | 0.8446 | 0.8930 |
| | DCT policy | | | | | |
| Setting | Success Rate | SVDDED | TaskEmbED | LSTMED | DCTED | PrObe |
| all optimal | $80.56 \pm 11.65\%$ | 0.3789 | 0.3597 | 0.4468 | 0.5734 | 0.7474 |
| 3 optimal & 2 sub | $78.61 \pm 14.02\%$ | 0.3908 | 0.3811 | 0.4606 | 0.6019 | 0.7530 |
| all sub-optimal | $77.22 \pm 14.16\%$ | 0.3899 | 0.4051 | 0.4718 | 0.5341 | 0.7588 |
| | SCAN policy | | | | | |
| Setting | Success Rate | SVDDED | TaskEmbED | LSTMED | DCTED | PrObe |
| all optimal | $75.56 \pm 12.68\%$ | 0.4653 | 0.4657 | 0.4487 | 0.6757 | 0.7505 |
| 3 optimal & 2 sub | $71.67 \pm 11.18\%$ | 0.4985 | 0.5165 | 0.4979 | 0.7467 | 0.7804 |
| all sub-optimal | $69.72 \pm 12.30\%$ | 0.4886 | 0.5262 | 0.5096 | 0.7140 | 0.7752 |

**Demonstration quality**    We verify the influence of demonstration quality on both the FSI policy and AED methods, specifically testing it on the *Press Button* task. We collected sub-optimal demonstrations that failed to press the button on the first trial but successfully pressed it on the second try. Then, the policies and AED methods used these demonstrations to perform the task.

We expected the policy performance to decrease as the demonstration quality decreases. Therefore, AED methods should achieve a higher AUPRC score if they are not affected by the quality changes of demonstrations, as it becomes easier to detect errors. However, from Table 7, we observe that not all AED methods obtain higher scores, indicating that they are also affected by the decrease in quality.

Notably, our proposed PrObe is the only method that achieves better results for all cases when comparing the first and second rows, and the first and third rows for each policy. Additionally, an interesting observation can be found in the case of NaiveDC policy (comparing all optimal vs. all sub-optimal): LSTMED and TaskEmbED obtained similar AUPRC scores, as they do not access the demonstration information directly; SVDDED suffered in this situation since it averages every demonstration frames (including those sub-optimal movements); DCTED and our PrObe achieved slightly better results because they filter useful information from the demonstrations.

Table 8: Viewpoint changes. **AUROC** [↑] is reported. We study the impact of viewpoint changes on AED methods' performance. The camera is slightly shifted away from the table during inference, which makes detecting errors more difficult. Nevertheless, our PrObe is the least affected method.

| Viewpoint | SVDDED | TaskEmbED | LSTMED | DCTED | PrObe |
|---|---|---|---|---|---|
| original | 0.5694 | 0.5465 | 0.4929 | 0.7501 | 0.7498 |
| shift | 0.4821 | 0.4959 | 0.4667 | 0.7056 | 0.7225 |
| difference | -15.34% | -9.27% | -5.32% | -5.93% | **-3.64**% |

**Viewpoint changes**    As AED methods detects behavior errors through visual observations, changes in camera viewpoint will affect the ease of observing errors. We conducted the experiment in the *Press Button* task with the DCT policy and all AED methods. In this experiment, the camera is slightly shifted away from the table and robot arm, which makes detecting errors more difficult. Besides, the policy is also impacted because the policy and AED methods utilize the same camera. Thus, the performance of DCT policy decreases from $80.00 \pm 10.80\%$ to $78.33 \pm 11.18\%$.

From Table 8, although the policy yields more behavior errors after camera shifted, the performance of all AED methods decreases due to the increased difficulty of detecting errors. Nevertheless, our PrObe is the least affected method, which demonstrates its superiority against other approaches.

## F    Limitations

**Offline evaluation in AED inference?**    Our ultimate goal is to terminate (or pause) the policy instantly when detecting a behavior error online. However, in our main experiment, only the trained policies performed in novel environments online, and their rollouts were collected. AED methods later use the rollouts to predict errors for a fair comparison. Nevertheless, we illustrate that PrObe effectively address the online detection challenge through the time accuracy experiment. Additionally, the pilot study on error correction involves running AED methods online to promptly correct errors.

**No comparison with state-of-the-art methods?**    While there are no available state-of-the-art (SOTA) methods for the novel AED task we formulate, we have endeavored to design several strong baselines with various characteristics (cf. Section C of the Appendix). These baselines draw inspiration from relevant areas such as one-class classification, metric-learning, temporal modeling. Moreover, their number of learnable parameters is ten times that of our PrObe. Nonetheless, PrObe demonstrates superior performance compared to them in our extensive experiments.

Additionally, we observed a parallel line of research focused on detecting or correcting robot errors, with some SOTA approaches [51, 49, 50, 56]. However, our approach and theirs have different natures in terms of the task scenario. Their methods typically rely on human-in-the-loop guidance, whereas in our setting, the robot operates in an unseen environment, and the expert may only be able to provide demonstrations without the capability to correct the robot's failures. Consequently, they are not suitable as baselines in this work.

**Not evaluated on benchmarks or in real-world environments?**    Due to the practical challenges inherent in FSI tasks and the AED task, we have had to develop our own benchmarks and tasks. Moreover, the lack of real-world experiments is attributed to the following reasons: (1) Deploying the necessary resources for the AED task in the real world is high-cost and requires a long-term plan due to safety concerns. (2) The FSI policies still struggle to perform complex tasks in the real world, making assessments less than comprehensive. (3) Although PrObe achieves the best performance, there is still significant room for improvement in addressing the AED task. We emphasize that hastily conducting AED and FSI research in real environments may cause irreparable damage.

Even previous error correction work [19] or related safe exploration research [57, 58, 59] was conducted in simulation environments, demonstrating that it is reasonable and valuable for us to conduct detailed verification in simulations first. As mentioned earlier, we have developed seven challenging and realistic multi-stage FSI tasks, each containing dozens of base and novel environments. To our knowledge, the simulation environments we established are the closest to real-world scenarios in the literature.

## Broader Impact

Our work focuses on addressing a critical learning task, specifically the error detection of policy behaviors, with the goal of accelerating the development of safety-aware robotic applications in real-world scenarios. While we have not identified any immediate negative impacts or ethical concerns, it is essential for us to remain vigilant and continuously assess potential societal implications as our research progresses.

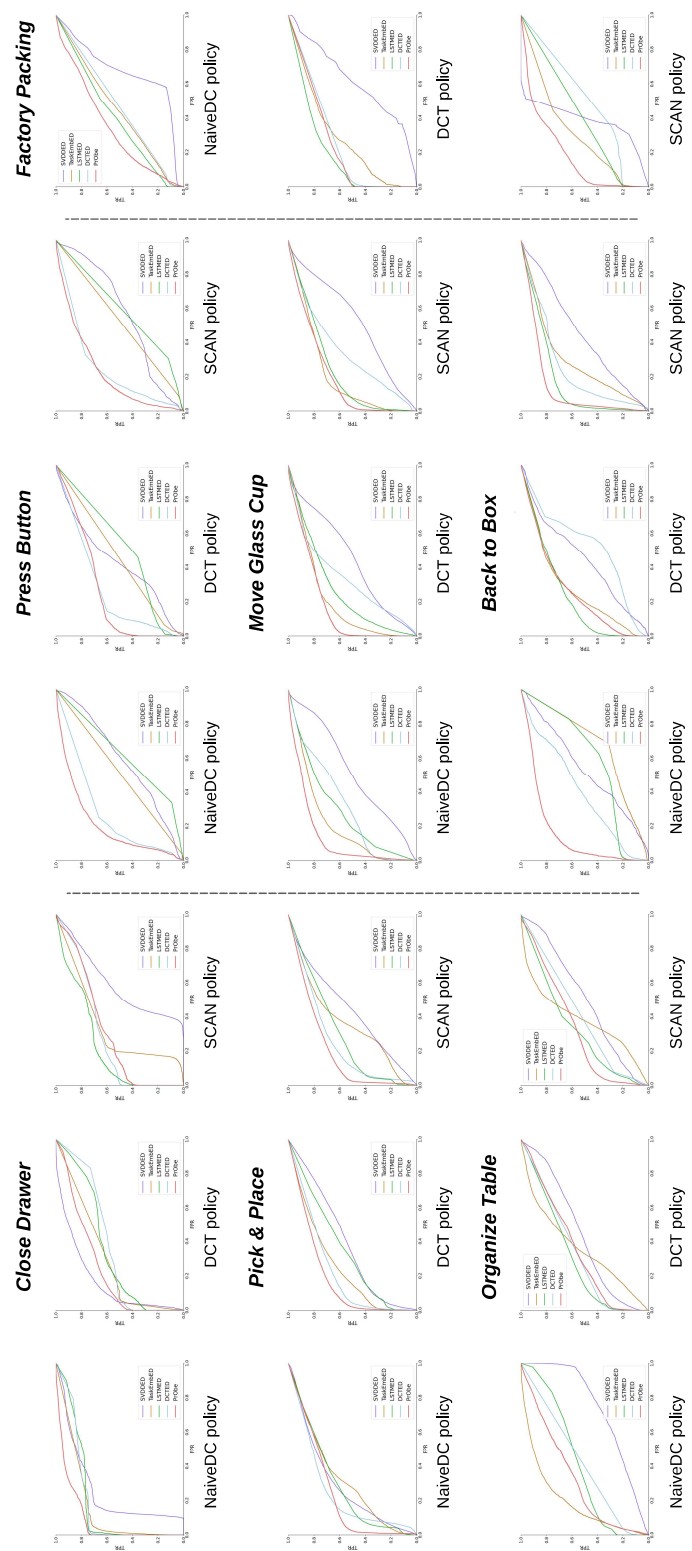

Figure 15: AED methods' ROC curves for all FSI tasks. The AUROC scores reported in the main paper's Figure 4 are the area under the curves.

