# OpenReview forum: "AED: Adaptable Error Detection for Few-shot Imitation Policy"
_NeurIPS.cc/2024/Conference — NeurIPS 2024 poster_

### Official Review · Reviewer_SywQ · 2024-06-13

**Soundness:** 3
**Presentation:** 3
**Contribution:** 3
**Rating:** 7
**Confidence:** 3

**Summary:**

The paper aims to address the adaptable error detection problem, i.e., detecting the error behavior of a robot policy in an unseen environment. Solving the problem ensures that the robot is stopped before performing behavior that causes disruptions to the surroundings.

The AED problem introduces three challenges: the abnormal behavior in the unseen environment is not observed in the training data and common anomaly detection method cannot be used; with complex background, there is no noticeable change between frames and this makes it difficult to indicate when the error occurs; AED is required to terminate the policy timely once the error occurs. The three challenges make it difficult to apply the current approaches on anormaly detection.

The paper creates a new benchmark specially designed for adaptable error detection, which consists of 322 base and 153 new environments encompassing six indoor tasks and one factory task.

The paper proposes a PrObe architecture consists of a pattern extractor and a flow generation equipped with a pattern loss, a temporal contrastive loss and a classification loss to predict the error probability.

The paper conducts experiments on the proposed benchmark and achieves the top 1 counts, average ranking, and average performance difference. Extensive ablative study justifies the effectiveness of the design choices.

**Strengths:**

Few-shot imitation learning learns a robot policy for a new environment with only a few demonstrations, which could accelerates the development of robot applications. However, with only a few demonstrations, there exist many corner cases that the robot could not observe in the demonstrations and the appearance and background of the scene are also different. This makes the policy perform error bahavior, which makes the surroundings fall into a dangerous situation. Thus, the problem of adaptable error detection is important to solve.

The three loss designs not only supervisedly learn the error prediction but also unsupervisedly regulate the intermediate feature to follow some human priors including close features for frames with high task relation and sparse feature to detect frame changes.

The experiment datasets and tasks are specially designed for the AED problems and selected baselines are strong error detection baselines.

**Weaknesses:**

One main contribution of the paper is that error detection could avoid the robots making disrupted behavior in the real-world environment. Thus, the authors may need a real-world environment that contains some disruption scenarios and test the method in such environments. Even the most realistic simulation may not simulate some real factors such as inaccurate control program of the robot arm.

**Questions:**

On line 207-208, normalizing by L2 norm could make all the features have the same norm but how could this mitigate the biases? Even the unit vector may only learn the background information in the visual inputs.

On line 212-215, the authors need to note that the task embeddings are extracted using prior works, which is introduced only in the experiments.

Could the authors provide more details or some examples on how to collect the positive and negative pairs for L_{tem}?

---

> ### Author Rebuttal · Authors · 2024-08-05
>
> Dear Reviewer SywQ,
>
> Thank you for your insightful comments and high recognition of our work! Our responses below are dedicated to addressing your questions and concerns.
>
> ---
>
> **[W1] The authors may need a real-world environment that contains some disruption scenarios and test the method in such environments.**
>
> - Thank you for the suggestion. As we discussed in the Limitation section (lines 745-758), there are still several obstacles to performing our AED task in real-world environments. Considering the capabilities and safety concerns of few-shot imitation (FSI) policies and AED methods in real-world environments, we first verified the AED tasks in realistic simulated environments. This approach was also inspired by previous work [19] and related research fields (e.g., safe exploration [57, 58, 59]).
> - We will continue to advance the progress of our AED task, including future sim-to-real and real-world experiments. Thank you again.
>
> **[Q1] On line 207-208, normalizing by L2 norm could make all the features have the same norm but how could this mitigate the biases?**
>
> - According to our experience, the feature values extracted from observations in different environments vary greatly, and such deviations may exacerbate errors in model judgment. Therefore, we divide the extracted features by their L2 norm to turn them into unit vectors, and let the subsequent pattern observer learn on the unit vectors. This can eliminate the deviation in value scale caused by domain (environment) changes. However, unit vectors from different domains may still have angular deviations, so in lines 207-208 we use the term "mitigate" rather than "eliminate."
>
> - We will continue to explore advanced strategies in domain adaptation to further mitigate the effects of domain change on the model. Thank you for this question.
>
> **[Q2] On line 212-215, the authors need to note that the task embeddings are extracted using prior works, which is introduced only in the experiments.**
>
> - Thank you for the suggestion. As you mentioned, these task embeddings are extracted from various few-shot imitation (FSI) policies, so how they are extracted can only be introduced when discussing which FSI policies are ultimately used. However, regarding the function of task embeddings (demonstrations) in our AED task (also mentioned by Reviewer qV6D), we can indeed provide more explanation in Section 5.2.
> - Please check our response to Reviewer qV6D's W2 for more details. Thank you again.
>
> **[Q3] Could the authors provide more details or some examples on how to collect the positive and negative pairs for $L_{tem}$?**
>
> - Thank you for the question. First, please note that according to our setting, the failed rollouts $X^b_{fail}$ in the base environment have frame-level labels. For each sampled failed rollout in each training iteration, we first randomly select one frame as the anchor. The anchor may be a normal frame or a frame where the error has occurred. Based on the selected anchor, we will sample the positive and negative samples using the frame-level labels of the failed rollout. After selecting the three samples, we calculate the temporal distance between the anchor and the positive sample, and the anchor and the negative sample. Then, we adjust the temporal-aware margin in $L_{tem}$  based on these temporal distances (line 230). Through this process, our $L_{tem}$ objective can simultaneously consider the latent and temporal correlations between frames.
>
> - We will include these details in our final version. Thank you again for this question.
>
> ---
>
> - Thank you again for your time and dedication in reviewing our paper. We appreciate your positive assessment of our work.
> - If you have any unresolved concerns after reading our responses, please let us know. We look forward to learning more from you during our constructive discussions.

---

> ### Author Response · Authors · 2024-08-12
>
> Dear Reviewer SywQ,
>
> Thank you once again for your valuable reviews and comments. We have provided additional details to address the concerns you raised.
>
> Since the discussion period  **ends in two days**, we would appreciate knowing if there are any unresolved issues that require further clarification. If so, we would be happy to discuss them with you. Otherwise, if all your concerns have been addressed, we kindly ask you to consider adjusting your evaluation score accordingly.
>
> Thank you again for your time and consideration.

---

### Official Review · Reviewer_3VgP · 2024-06-30

**Soundness:** 4
**Presentation:** 4
**Contribution:** 3
**Rating:** 7
**Confidence:** 4

**Summary:**

This paper introduces Adaptable Error Detection (**AED**) within Few-Shot Imitation (**FSI**) tasks, a critical yet underexplored area in robotics and AI. The authors establish a novel benchmark for assessing AED methods and present **PrObe**, designed specifically for this task. Through comprehensive evaluations, the paper demonstrates that PrObe outperforms baseline models and shows robustness across different FSI policies.

**Strengths:**

**S1. Well-written.** The paper has a coherent narrative and logical flow. The problem scenario is effectively motivated with real-life examples that illustrate the practical importance and applicability of the research. Relevant literature is extensively discussed and consistently referenced throughout the paper.

**S2. Illustrations.** The schematic illustrations provided in the paper are well made and enhance the comprehensibility of AED, PrObe, and their experimental analysis. The figures effectively delineate the structured protocol of the AED task across different phases, and the intricate architecture of PrObe, showcasing its components and their functions.

**S3. Unique challenges.** The unique challenges established by AED distinguish it from FSAD and other tasks in prior works. AED's focus on online anomaly detection in novel environments, where behavioral errors occur without clear visual cues, establishes it as a critical area of research.

**S4. Novel components.** PrObe establishes the usefulness of its design principles by outperforming other baselines. Augmenting rollouts, extracting patterns from policy embeddings, generating a pattern flow and fusing it with task-embeddings, and utilizing a temporal-aware triplet loss - all contribute to its effectiveness.

**S5. Comprehensive analysis.** The paper extensively explores PrObe's application in FSI policies for AED. PrObe achieves better visual separability of features in the latent space between successful and erroneous trajectories, thereby enhancing the interpretability of learned embeddings. Notably, PrObe excels in temporally precise error detection, outperforming other methods that detect errors too late or too early. An in-depth ablation study confirms the stability and essential contributions of PrObe’s components. Additionally, the paper explores PrObe’s failure scenarios, the impact of demonstration quality, and viewpoint changes.

**Weaknesses:**

W1. While the textual descriptions in the Preliminaries section provide a solid understanding of the DC policy, incorporating mathematical notation would significantly enhance clarity and comprehension. Specifically, the paper could benefit from the following mathematical formulations:
 1. The contents of history $h$, detailing what it encapsulates
 2. The feature encoder and task-embedding network computing the history, demonstration, and task-embedding features
 3. Task-embedding network padding the sequences
 4. The actor policy, e.g, $\pi(a|o_t, f_h, f_{\zeta})$
 5. The inverse dynamics module
 6. NLL and MSE objective in this particular setting

W2. The average difference metric presented in Figure 4, which compares each method's performance to the second-best performing method, may be inadequate. This approach is only sensible when comparing two methods. A more informative assessment might measure performance gains relative to the worst-performing method, or performance losses relative to the best-performing method. Alternatively, using a standardized baseline or oracle for comparisons could provide a clearer and more meaningful evaluation.

W3. PrObe is only shown to be effective for evaluating policies trained on image data. As a primary example, the authors motivate the FSI ED problem setting on robotic tasks. Although sensor data cannot always fully capture the properties and locations of separate objects, policies for robotic manipulation tasks are often trained on proprioceptive data—such as joint angles and torques—to achieve better precision. Image data, though rich in environmental context, can sometimes be unavailable or impractical due to factors like occlusions, lighting conditions, and higher training costs. The inherent differences between image-based and proprioceptive data mean that the latent space characteristics and the patterns critical for error detection would vary. Image-based models capture visual features like shapes and spatial relationships, whereas proprioceptive-focused models emphasize dynamics and kinematic features, including the robot’s mechanical interactions. While there's no inherent reason to doubt PrObe’s capability with different data modalities, the effectiveness of its pattern extractor when applied to latent features learned from proprioceptive data remains uncertain without empirical validation. Conducting experiments with proprioceptive sensor data and demonstrating successful results would further validate the robustness, generality, and applicability of PrObe.

### Minor Improvements

1. The second and third contributions—introducing a novel benchmark and PrObe—are blended together; these could be separate for clarity.
2. Please add a reference in the main paper that the limitations are in Appendix F.
3. Line 125 “semantically” refers to language or linguistics, which doesn’t seem to be the intended use of the word here
4. Line 133 “learns implicit mapping” → “learns an implicit mapping”
5. Line 136 “current history” → “the current history”
6. Line 154 “and few” → “and a few”
7. Line 168 “high controllable” → “highly controllable”
8. Line 174 “and few” → “and a few”
9. Line 192 “from agent’s” → “from the agent’s”
10. Line 205 “take as input the history features $f^h$” →“takes the history features $f^h$ from … as input”
11. Footnote on page 4 “thet” →”they”
12. Line 614 “are are” → “are”
13. Line 661 “*Pick & Plcae*” → “*Pick & Place*”

**Questions:**

Q1. Why did the authors choose the main conference track instead of the Datasets & Benchmarks track if they propose a benchmark?
Q2. What is the intervention method employed at critical moments to the agent’s policy to collect failed rollouts?
Q3. Why were TaskEmbED and LSTMED chosen among the 4 baselines for embedding visualization?
Q4. Is AED only applicable for error detection on demonstration-conditioned policies or does it extend to more general policies?
Q5. Is PrObe also able to evaluate policies that have been trained for robotic manipulation on proprioceptive data, not image data?

**Limitations:**

The authors have extensively discussed the limitations of their work.

---

> ### Author Rebuttal · Authors · 2024-08-05
>
> Dear Reviewer 3VqP,
>
> Thank you for your insightful comments and high recognition of our work! Our responses below are dedicated to addressing your questions and concerns.
>
> ---
>
> **[W1] Incorporating mathematical notation would enhance clarity and comprehension.**
>
> - Thank you for your suggestion. We will add mathematical notations to the cases you listed. For example, we will use $h_t \coloneqq (o_0, o_1, \ldots, o_t)$ for the rollout history at time $t$, where $o$ represents the observation. We will also review the paper to identify any parts that can be enhanced for clarity and comprehension.
>
> **[W2] The average difference metric presented in Figure 4 may be inadequate.**
>
> - Thank you for pointing out this important issue. As per your suggestion, we provide the average performance difference metric comparing each method with the **worst** and **best** methods in the following tables.
>
>     Comp. w/ the worst method:
>
>     |  | SVDDED | TaskEmbED | LSTMED | DCTED | PrObe |
>     | --- | --- | --- | --- | --- | --- |
>     | AUROC | 7.05% | 41.63% | 41.59% | 44.20% | 67.29% |
>     | AUPRC | 2.61% | 35.42% | 61.67% | 57.13% | 78.03% |
>
>      Comp. w/ the best method:
>
>     |  | SVDDED | TaskEmbED | LSTMED | DCTED | PrObe |
>     | --- | --- | --- | --- | --- | --- |
>     | AUROC | -34.87% | -16.90% | -15.95% | -15.44% | -2.45% |
>     | AUPRC | -32.28% | -19.63% | -10.87% | -13.17% | -1.06% |
> - From the results under these two new metrics, it's obvious that our PrObe method significantly outperforms the compared baselines. We will include these new results in our final version.
>
> **[W3, Q5] Is PrObe also able to evaluate policies trained on proprioceptive data?**
>
> - Thank you for the interesting question. We agree that it is worth exploring whether PrObe can observe pattern changes in proprioceptive data (data in another modality). However, considering our work's setting (refer to the scenario presented in Section 4), it may not be feasible to conduct experiments on proprioceptive data for two main reasons:
>
> 1. The configurations of experts and agents are different. It may be very challenging for few-shot imitation (FSI) policies to learn tasks from different joint sets. As you mentioned, the proprioceptive data may not provide all the necessary task-related information that could potentially be extracted from image data.
> 2. Our setting assumes that we have less information and control in novel environments. Even if we assume that the error detector and FSI policy can be integrated into the same agent, we may not be able to access the expert's proprioceptive data (for example, if the expert is a customer).
>
> - Although it is difficult to conduct experiments on training PrObe with proprioceptive data under our current problem setting, we will explore whether PrObe has the ability to learn from data of different modalities in the future.
>
> **[Q1] Why did the authors choose the main conference track instead of the Datasets & Benchmarks track?**
>
> - Thank you for the question. In this work, we regard the AED task as a new learning problem. In addition, we propose a new method, PrObe, to address this challenging task. The AED benchmark is a platform to verify the effectiveness of various AED methods. For us, the first two parts (a new task and a new method) are the main contributions of this work, and thus, we decide to submit it to the main track.
>
> **[Q2] What is the intervention method employed at critical moments to collect failed rollouts?**
> - Thank you for the question. Different intervention methods are designed for different tasks, which can be roughly divided into two categories: (1) The interactive object is correct, but the interaction fails (e.g., failure to grasp the object, failure to press the button, failure to close the drawer), and (2) Interacting with the wrong objects (e.g., placing objects in the wrong container, closing the wrong drawer).
> - For the first case, we add a random shift to the accurate pose. For the second case, we replace the accurate pose with the pose of the incorrect object, which also includes a random shift. We will include these details in each task's description.
>
> **[Q3] Why were TaskEmbED and LSTMED chosen for embedding visualization?**
>
> - Due to space constraints when drafting the paper, we  report two baselines with the most different attributes in our embedding visualization experiment (cf. Table 3 in the Appendix). However, we can provide visualization results for all baselines. Please refer to the PDF file in our global rebuttal for more details. We will integrate all visualization results into our final version.
>
> **[Q4] Is AED only applicable for error detection on demonstration-conditioned policies?**
>
> - Thank you for the question. In the problem setting we are considering (recall the scenario provided in Section 4), the policy has to learn the task through observing demonstrations performed in a novel environment. Mainstream few-shot imitation policies following this setting are usually demonstration-conditioned, so our AED task and Probe method focus on detecting the errors caused by this type of policy.
> - If the policy can now be directly trained and tested in a single scene, e.g., conventional imitation learning, then intuitively, our Probe can work in this setting by simply removing the fusion process of pattern flow and task embeddings. In other words, Probe would use only the pattern flow to detect errors. We will continue to explore how to improve the versatility of Probe for various policies.
>
> **Typos & minor improvements**
>
> - Thank you so much for your careful review of our paper. We will fix these typos and revise the paper following your instructions.
> ---
> - Thank you again for your time and dedication in reviewing our paper. We appreciate your positive assessment of our work.
> - If you have any unresolved concerns after reading our response, please let us know. We look forward to learning more from you during our constructive discussions.

---

> > ### Comment · Reviewer_3VgP · 2024-08-09
> >
> > I appreciate the authors’ efforts in addressing my concerns and answering the questions.
> >
> > **W2**. I do think the results that you provided in this format are more coherent. I suggest you select one of them and swap out the current one in Figure 4. I would say that including both is unnecessary as they infer the same thing and it would just take up more space. I think the comparison w/ the worst method is more informative, as the scale is larger.
> >
> > **W3**. Thank you for the explanation. According to your reasoning, it indeed seems complicated to incorporate proprioceptive data. I do believe that camera footage for error detection is appropriate and sufficient, and image-based learning, after all, generally poses a greater challenge.
> >
> > I thank the authors for their work. All my concerns and inquiries have been addressed. I will keep the current score. A higher score would necessitate a more high-impact paper 1) tackling a broader setting, 2) surpassing prior methods by a landslide, or 3) having experiments conducted in many simulation environments or even physical robots.

---

> > > ### Author Response · Authors · 2024-08-10
> > >
> > > We are grateful for the reviewer's prompt and detailed feedback, which provided valuable advice during our discussion.
> > >
> > > We are also pleased that our responses have addressed all of the reviewer's concerns and questions. We would like to thank the reviewer once again for their high recognition of our work.

---

### Official Review · Reviewer_qV6D · 2024-07-13

**Soundness:** 3
**Presentation:** 2
**Contribution:** 3
**Rating:** 7
**Confidence:** 3

**Summary:**

This paper proposed a task Adaptable Error Detection (AED) that attempts to perform online behavior error detection for FSI policies, it advocate three main challenges comparing to FSAD: novel environment, no notable change for behavior error that AED tries to detect, and it has to be conduct simultaneously with policy execution to ensure timely error detection and damage control. The AED benchmark is provided, and a AED method, PrObe is proposed to solve AED through error prediction based on fusion of task embedding and flow pattern generated from history of rollout. The PrObe achieve good performance compare to baselines, and paper provided comprehensive studies analyzing PrObe error detection timing, it's embedding pattern, the importance of components, the stability of error detection, and provided pilot error correction to demonstrate practicality of the AED task.

**Strengths:**

1. AED is a important task for FSI research.
2. The proposed PrObe is very effective in AED task for various tasks.
3. abundant analysis is provided on PrObe timing, embedding analysis, performance stability, ablation and more.

**Weaknesses:**

The writing has improvement room:
1. line 161 - line 162: I am confused: if frame level label is not available, how come the measurement is averaging over actions?
2. The function of demonstration data is not clearly described in section 5.2, more explanation is helpful.

The AED benchmark is not well described in the writing.

**Questions:**

See above.

**Limitations:**

The limitation in terms of proposed PrObe is discussed through analysis experiments, which make sense to me.

---

> ### Author Rebuttal · Authors · 2024-08-05
>
> Dear Reviewer qV6D,
>
> Thank you for your insightful comments and high recognition of our work! Our responses below are dedicated to addressing your questions and concerns.
>
> ---
>
> **[W1] If frame-level label is not available, how come the measurement is averaging over actions?**
>
> - We try to make Eq. 1 a general form, so we express it in such a way that we can obtain detection accuracy across *frames*. However, as we illustrated in the practical scenario (lines 163-166), frame-level labels are often unavailable during the testing phase in practice. Therefore, the A($\cdot, \cdot$) we use in testing is sequence-level. In addition, we conduct the timing accuracy experiments to verify the accuracy of different methods in determining the timing of error occurrences.
> - We will revise this part to make it more clear. Thanks for the question.
>
> **[W2] The function of demonstration data is not clearly described in section 5.2.**
>
> - The demonstration data originally served as important information for the few-shot imitation policy to understand how to perform tasks in novel environments. In our newly proposed AED task, these demonstrations are also used for AED methods to determine whether the current state deviates from the demonstrations.
> - We agree with the reviewer that we can provide more explanation on the function of demonstration data in our AED task and will include it in our final version. Thank you for pointing out this issue.
>
> **[W3] The AED benchmark is not well described in the writing.**
> - We have provided the details of our AED benchmark in **Section D of the Appendix**, including the task attributes, each task's description, success conditions, and potential behavior errors. We also refer readers to Section D of the Appendix when introducing evaluation tasks (Section 6.1, line 246) in the main text.
> - If there are any additional details about our AED benchmark that the reviewer is interested in, please feel free to let us know.
>
> ---
>
> - Thank you again for your time and dedication in reviewing our paper. We appreciate your positive assessment of our work.
> - If you have any unresolved concerns after reading our responses, please let us know. We look forward to learning more from you during our constructive discussions.

---

> ### Author Response · Authors · 2024-08-12
>
> Dear Reviewer qV6D,
>
> Thank you once again for your valuable reviews and comments. We have provided additional details to address the concerns you raised.
>
> Since the discussion period  **ends in two days**, we would appreciate knowing if there are any unresolved issues that require further clarification. If so, we would be happy to discuss them with you. Otherwise, if all your concerns have been addressed, we kindly ask you to consider adjusting your evaluation score accordingly.
>
> Thank you again for your time and consideration.

---

> > ### Comment · Reviewer_qV6D · 2024-08-13
> >
> > Thanks for addressing my questions, i am happy to increase my score to 7.

---

> > > ### Author Response · Authors · 2024-08-13
> > >
> > > Thank you for your feedback! We are pleased that our responses have addressed your concerns. We will revise the paper following your valuable suggestions.
> > >
> > > We are also grateful to the reviewers for their support and for increasing their ratings of our work. Thank you again.

---

### Author Rebuttal · Authors · 2024-08-05

We would like to thank all reviewers for their constructive and insightful comments on our work.

We are excited that our work possesses several strengths recognized by the reviewers, as summarized below:

- Our proposed AED task is **important** (all reviewers) with **unique challenges** (3VgP).
- Our proposed solution PrObe contains **novel components** (3VgP) and is **effective** (all reviewers).
- Our work provided comprehensive analysis (qV6D, 3VgP); the compared **baselines** are **strong** (SywQ).
- Our paper is **well-written** (3VgP) with **well-made illustrations** (3VgP).

Regarding the concerns and questions raised by the reviewers, we have provided **more details** and **evaluation results under new metrics** to address them. Additionally, we have analyzed the feasibility of conducting the studies suggested by the reviewers. Finally, we will refine our paper based on the improvement suggestions from all reviewers.

We have posted our responses under each reviewer's comment separately. Please let us know if there are still any unresolved concerns.

Thank you again for reviewing our paper.

---

In response to the reviewer's inquiry, the attached PDF provides embedded visualizations of all baselines. Please check the explanation in the PDF with Table 3 in the Appendix.

---

### Decision · Program_Chairs · 2024-09-25

**Decision:**

Accept (poster)

**Comment:**

This paper is about a new task, adaptable error detection, in context of few-shot imitation policy learning, where the task is to detect behavior errors made by the few shot imitation policy. from visual observations in unseen environments.

The paper proposes a new task (AED), an associated benchmark, and a novel model to solve the AED task called PrObe.

All reviewers agree that the paper proposes a novel task (AED) and approach (PrObe), the AED task is very relevant for few-shot imitation learning, it introduces new challenges, and PrObe contains novel components. The paper is well written, and evaluation seems correct. This AC agrees with these evaluations.

The paper needs some minor improvements, mainly to clarify benchmark details, and possibly shuffle some details between supplementary and main paper, to make sure the important details are in the main paper, and the mathematical descriptions of tasks and models. The rebuttal and discussion provide detailed feedback to improve the paper.

Overall this paper is considerably above the acceptance threshold, and should be accepted.